# Extremely stable graphene electrodes doped with macromolecular acid

Sung-Joo Kwon[1], Tae-Hee Han[2,3], Taeg Yeoung Ko[4], Nannan Li [5], Youngsoo Kim[6], Dong Jin Kim[6,7], Sang-Hoon Bae[3], Yang Yang[3], Byung Hee Hong [7], Kwang S. Kim [5], Sunmin Ryu[4] & Tae-Woo Lee[2,8]

Although conventional p-type doping using small molecules on graphene decreases its sheet resistance ($R_{sh}$), it increases after exposure to ambient conditions, and this problem has been considered as the biggest impediment to practical application of graphene electrodes. Here, we report an extremely stable graphene electrode doped with macromolecular acid (perfluorinated polymeric sulfonic acid (PFSA)) as a p-type dopant. The PFSA doping on graphene provides not only ultra-high ambient stability for a very long time (> 64 days) but also high chemical/thermal stability, which have been unattainable by doping with conventional small-molecules. PFSA doping also greatly increases the surface potential (~0.8 eV) of graphene, and reduces its $R_{sh}$ by ~56%, which is very important for practical applications. High-efficiency phosphorescent organic light-emitting diodes are fabricated with the PFSA-doped graphene anode (~98.5 cd A$^{-1}$ without out-coupling structures). This work lays a solid platform for practical application of thermally-/chemically-/air-stable graphene electrodes in various optoelectronic devices.

[1] Department of Materials Science and Engineering, Pohang University of Science and Technology (POSTECH), 77 Cheongam-Ro, Nam-Gu, Pohang, Gyungbuk 37673, Republic of Korea. [2] Department of Materials Science and Engineering, Seoul National University, 1 Gwanak-ro Gwanak-gu, Seoul 08826, Republic of Korea. [3] Department of Materials Science and Engineering, California NanoSystems Institute, University of California, Los Angeles, CA 90095, USA. [4] Department of Chemistry and Division of Advanced Materials Science, Pohang University of Science and Technology (POSTECH), 77 Cheongam-Ro, Nam-Gu, Pohang, Gyungbuk 37673, Republic of Korea. [5] Department of Chemistry, Center for Superfunctional Materials, Ulsan National Institute of Science and Technology (UNIST), Ulsan 44919, Korea. [6] Graphene Square Inc., Inter-University Semiconductor Research Centre, Seoul National University, Seoul 08826, Korea. [7] Program in Nano Science and Technology, Graduate School of Convergence Science and Technology, Seoul National University, 1 Gwanak-ro Gwanak-gu, Seoul 08826, Republic of Korea. [8] Nano Systems Institute (NSI), Institute of Engineering Research, Research Institute of Advanced Materials, Seoul National University, 1 Gwanak-ro Gwanak-gu, Seoul 08826, Republic of Korea. These authors contributed equally: Sung-Joo Kwon, Tae-Hee Han. Correspondence and requests for materials should be addressed to T.-W.L. (email: twlees@snu.ac.kr)

Graphene has outstanding electrical, mechanical, and optical properties[1-5], so it has been regarded as an alternative to indium tin oxide (ITO), which is the conventional transparent electrode in optoelectronic devices but is not suitable for electrodes in flexible optoelectronics due to its brittleness and increasing cost[6,7]. Since the development of chemical vapor deposition (CVD) methods to produce high-quality and large-area graphene[2-5], much research has been devoted to applying graphene in flexible electronics such as organic light-emitting diodes (OLEDs)[8-16], organic solar cells[17-19], and organic transistors[20-22]. However, pristine graphene has high sheet resistance ($R_{sh} > 300\,\Omega\,sq^{-1}$) and low work function (WF ~4.4 eV) which are still inferior to those of ITO ($R_{sh}$ ~10 $\Omega\,sq^{-1}$, WF ~4.8 eV), so pristine graphene must be modified before it can be a practical replacement for ITO electrodes[23,24].

Various researchers had used chemical doping to control the electrical properties of pristine graphene[3,8,15,25-37]. Dopants used in graphene for electrodes to date have been mainly classified into two types: (1) small molecules[3,8,25-35], and (2) transition metal oxides[14,36,37]; both exploit charge transfer on the graphene surface. Charge-transfer doping of graphene with small-molecule dopants such as inorganic small-molecule acids (e.g., $HNO_3$, HCl, $H_2SO_4$)[3,8,25] and metal chlorides (e.g., $AuCl_3$, $FeCl_3$)[8,26-29] has been widely used and developed to increase the electrical conductivity of graphene; however, graphene that is doped with inorganic small-molecule acid has serious environmental instability, which has been considered as a great impediment to practical application of graphene electrodes. $R_{sh}$ of an $HNO_3$-doped graphene gradually increases under ambient conditions due to high volatility of small-molecule acids, and high temperature accelerates significant degradation in its electrical conductivity[8]. After doping with metal chlorides, reduction of metal cations can produce metal particles on the graphene surface; these have two deleterious effects: they can decrease the optical transmittance (OT) of the graphene, and if they are large, they protrude and provide leakage paths for electrical current in thin-film devices[8,26-29]. Also, transition metal oxides are not uniformly deposited on the graphene surface because the graphene lacks dangling bonds or surface functional groups. Therefore, thermal evaporation of transition metal oxide on graphene also roughens its surface[38]. Ideal chemical doping of graphene for practical use as an anode in electronic devices should achieve: (1) low $R_{sh}$, (2) high WF, (3) high stability against heat, chemical, and ambient conditions, (4) smooth film surface, and (5) high OT.

Here, we introduce a novel approach to macromolecular chemical doping that uses a polymeric acid, which has not been used for graphene doping and demonstrated extraordinary results on all kinds of aspects of stability (high temperature, chemicals, and air) and high WF; these characteristics are almost unattainable with conventional small-molecule acid doping. We use a perfluorinated polymeric sulfonic acid (PFSA) for application to flexible graphene anodes in optoelectronics. PFSA is composed of a perfluorinated carbon backbone and sulfonic acid groups. Due to the electron-withdrawing properties from electronic dipole of acidic proton in sulfonic acid groups, PFSA induces p-type doping of graphene. Furthermore, the perfluorinated carbon backbones have large ionization potential, which substantially increases the surface potential of graphene. Superior thermal and chemical stability of PFSA molecule provided outstanding stability of chemically doped graphene against high temperature heating and exposure to various chemicals including strong acids and bases[39,40]. Therefore, the PFSA is an ideal form of macromolecular p-type dopant for graphene electrode. Our unconventional approach of using polymeric acid may stimulate research into high-stability graphene doping, and can be a starting point in the development of polymeric-acid dopants for ideal graphene electrodes.

## Results

**Doping effects of a polymeric fluorinated acid on graphene.** We used non-volatile polymeric acid (i.e., PFSA) instead of volatile small-molecule acid (e.g., $HNO_3$) to dope graphene (Fig. 1a). To study PFSA doping characteristics on graphene, we spin-cast a PFSA on high-quality and large-area single-layered graphene (SLG) that had been prepared using CVD on Cu foil, followed by conventional wet-transfer[4,8,10,14]. PFSA doping on graphene did not degrade the optical properties of graphene: at wavelength = 550 nm, pristine four-layered graphene (4LG) had OT = 90%, and PFSA-doped 4LG had OT = 89% (Fig. 1b).

Raman spectroscopy is a non-destructive tool to identify the doping characteristics, the number of layers, and the structural disorder in graphene[41,42]. Raman spectra were obtained from pristine and PFSA-doped SLG (Fig. 1c). The spectrum of the pristine graphene showed a very small D-band (~1350 $cm^{-1}$), which indicates that high-quality graphene had been successfully grown and transferred onto the substrate with few defects. The spectrum of PFSA-doped graphene also showed a very weak D band; this result proves that solution-processed doping of the PFSA does not induce any significant structural defects in the graphene lattice. The G- and 2D-bands of the PFSA-doped graphene were up-shifted compared to those of pristine graphene (G-band shift: from 1587 to 1594 $cm^{-1}$, 2D band shift: from 2684 to 2696 $cm^{-1}$); this change indicates that PFSA causes p-type doping of graphene[43].

To prove the uniformity of PFSA-doping over a large area, we prepared large area 4LG (> 3 cm × 4 cm) on glass substrate, then performed spatial mapping of $R_{sh}$ by using the EddyCus® TF map 2525SR system (Fig. 1d, e). Pristine 4LG without vacuum annealing showed $R_{sh} = 352.7 \pm 48.0\,\Omega\,sq^{-1}$. PFSA doping uniformly reduces the $R_{sh}$ of graphene throughout the large area, and PFSA-doping followed by 300 °C annealing showed significant reduction in the magnitude and variation in $R_{sh}$ (Pristine: 352.7 ± 48.0 $\Omega\,sq^{-1}$, PFSA-doped: 91.4 ± 30.1 $\Omega\,sq^{-1}$) (Fig. 1d, e). This result demonstrates the spatial large-area uniformity of PFSA-doped graphene.

The PFSA-doped graphene had a flatter and smoother surface (root mean square roughness ($R_{rms}$) ~0.495 nm) than that of pristine graphene ($R_{rms}$ ~1.47 nm) (Supplementary Fig. 1). Peak heights along the cross section of the PFSA-doped graphene were < 1 nm because deposition of thin polymeric layer can flatten uneven regions of the pristine graphene, such as wrinkles and grain boundaries. This result indicates that PFSA doping can yield a uniform surface of flexible graphene electrode without large particles.

**Thermal stability and doping mechanism.** The thermal stability of PFSA-doped and $HNO_3$-doped graphene samples was investigated. $HNO_3$ caused a stronger doping effect than PFSA: compared to pristine graphene, the $HNO_3$-doping reduced the $R_{sh}$ by ~69.7 ± 1.2%, which is more substantial than the reduction by PFSA doping (~52.4 ± 0.9%). To verify doping stability under high temperature, each p-type doped graphene (asD in Fig. 2a) was thermally annealed at 100 °C ≤ annealing temperature ($T_a$) ≤ 300 °C in ambient conditions.

High-temperature annealing affected $R_{sh}$ (Supplementary Table 1). Compared to $R_{sh}$ of the as-doped graphene, $R_{sh}$ of PFSA-doped graphene on Si/SiO$_2$ substrate gradually decreased as $T_a$ increased (up to ~22.4% decrease of $R_{sh}$ (asD) at $T_a$ = 300 °C), whereas $R_{sh}$ of $HNO_3$-doped graphene on Si/SiO$_2$ substrate

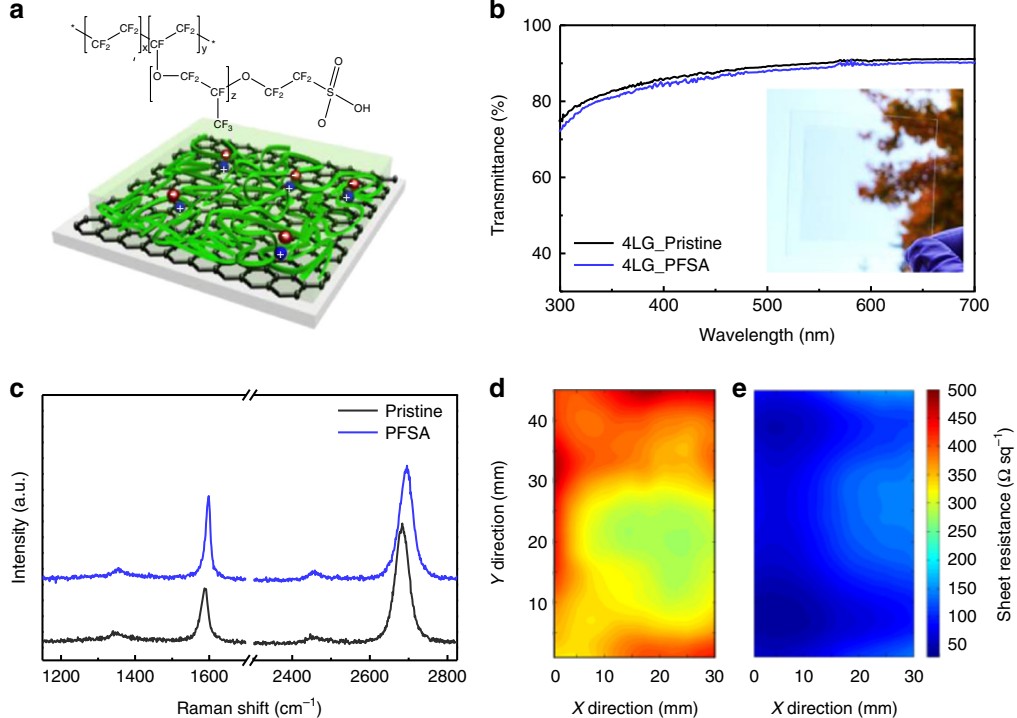

**Fig. 1** Macromolecular doping concept and its characteristics on graphene. **a** Chemical structure of perfluorinated polymeric sulfonic acid (PFSA), and schematic drawings of graphene doped using PFSA (+: hole, −: electron). **b** Optical transmittance of pristine and PFSA-doped 4LG (inset: large-area transferred 4LG on glass substrate). **c** Raman spectra of pristine and PFSA-doped graphene. Spatial $R_{sh}$ map of **d** pristine 4LG, and **e** PFSA-doped 4LG

increased as $T_a$ increased (~119.6% increase of $R_{sh}$ (asD) at $T_a$ = 300 °C) (Fig. 2a); this trend is due to the high volatility of HNO₃. These results indicate that thermal stability is much better in PFSA-doped graphene than in HNO₃-doped graphene.

Large-area mapping (> 3 cm × 4 cm) of $R_{sh}$ in the PFSA-doped graphene was performed after annealing at various $T_a$. Over the whole region, $R_{sh}$ decreased as $T_a$ of PFSA increased (Supplementary Fig. 2a−d). Potential difference after annealing at various $T_a$ was also identified using Kelvin probe measurement (Fig. 2b). The PFSA-doped graphene had a uniform WF over a large area (> 1.5 mm × 1.5 mm). PFSA-coating and annealing at 100 °C yielded ~0.73 eV increase of surface potential; this change is consistent with the results of ultraviolet photoelectron spectroscopy (UPS) (Supplementary Fig. 3). Thermal annealing also gradually increased the surface potential of graphene to ~1.01 eV at $T_a$ = 300 °C. The $R_{sh}$ decrease and WF increase after high-temperature annealing indicate that it increases p-type doping.

To determine the mechanism by which PFSA dopes graphene, we performed density-functional theory (DFT) calculation of PFSA-doped graphene. For ease of calculation, we used the simplest molecular form of PFSA. In the most energetically favorable configuration of PFSA-doped graphene (Fig. 2c, d and Supplementary Fig. 4), the acidic proton in the sulfonic acid group (−SO₃H) faces towards graphene. PFSA doping increases the WF of graphene by ~0.71 eV (Fig. 2c), which is consistent with results of Kelvin probe (~0.73 eV) and UPS (~0.8 eV). Acidic protons in sulfonic acid (−SO₃H) cause electronic dipole interaction with graphene, thereby reducing the electron density of graphene (Fig. 2d, green clouds). The substantial increase in WF can be attributed to the electronic dipole interaction of the acidic proton in sulfonic acid (−SO₃H) as well as an interface dipole formed by the electronegative perfluorinated backbone on

the graphene; this dipole caused by fluorinated substituents can induce a similar vacuum-level shift in self-assembled monolayers on the electrode surface[44–48]. The increased surface WF of the PFSA-doped graphene can effectively reduce the hole-injection energy barrier from the anode to overlying organic layers in OLEDs. Thermal annealing would cause rearrangement of PFSA at high $T_a$[49]. Thermal annealing causes the perfluorinated backbone that has low surface energy to become dominant at the film surface, and sulfonic acid groups (−SO₃H) in PFSA rearrange towards the underlying graphene. Therefore, high-temperature annealing of PFSA-doped graphene increases the p-type doping effect in PFSA-doped graphene.

To quantify the p-type doping effect in PFSA-doped graphene, we calculated the charge concentration ($n$) by using in-depth Raman spectrum analysis and the Dirac point shift in field-effect transistors (FETs). The hole-doping effect can be differentiated from the strain effect[50]. To monitor changes in $n$ and strain as a function of $T_a$ after PFSA doping, we obtained a hundred Raman spectra for each graphene sample by raster-scanning an area of 10 μm × 10 μm, then verified the variation of G-band and 2D-band position of each spectrum (Fig. 3a). G-band and 2D-band distributions of pristine graphene showed $n$ ~3 × $10^{12}$ cm$^{-2}$, which increased to ~8 × $10^{12}$ cm$^{-2}$ after PFSA doping without significant strain generated in the graphene layer. Without PFSA doping, thermal annealing did not increase $n$ in pristine graphene. However, PFSA-doped graphene caused gradual and considerable increase in $n$ as $T_a$ increased; this result indicates that increase of $n$ in PFSA-doped graphene can be attributed to rearrangement of PFSA at high $T_a$. Notably, PFSA-doped graphene treated at $T_a$ = 300 °C showed $n$ ~11 × $10^{12}$ cm$^{-2}$, which is three times higher than $n$ in pristine graphene (Fig. 3b, c). PFSA-doped graphene showed slight increase in strain (Pristine graphene: 0.1% to −0.2%, PFSA-doped graphene at

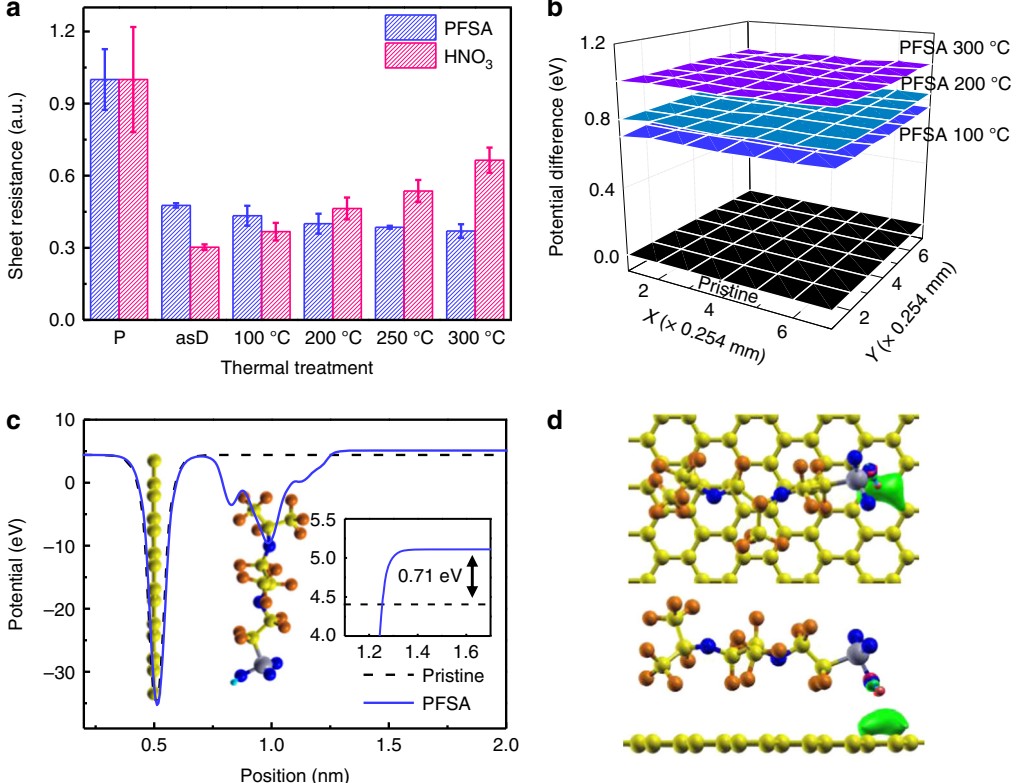

**Fig. 2** Temperature dependency and doping mechanism. **a** $R_{sh}$ changes of PFSA and $HNO_3$-doped graphene at various $T_a$. **b** Potential mapping of pristine, and PFSA-doped graphene annealed at various $T_a$. The error bars represent the s.d. of multiple measurement results. **c** Calculated electrostatic potential of the most stable configuration of PFSA-doped graphene (inset: difference in work function between pristine and PFSA-doped graphene). **d** Configuration of PFSA-doped graphene (top) top view, and (bottom) side view. Spheres: blue = oxygen, silver = sulfur, cyan = hydrogen, orange = fluorine, yellow = carbon. The isosurface of the charge density distribution difference is also shown in the figure. Isosurface level is 0.005e Å$^{-3}$

$T_a = 300\,°C$: 0.1% to −0.3%); this change can be attributed to slight thermal deformation during annealing.

We also fabricated graphene-based FETs on Si/SiO$_2$ (300 nm) substrate and measured electrical properties of FETs that used pristine or PFSA-doped graphene (Fig. 3d). FETs that used pristine graphene had a Dirac point of 6.2 ± 1.5 V. After PFSA doping and sequential thermal annealing, a Dirac point increased substantially to 106.2 ± 17.2 V; this change indicates that PFSA strongly p-doped graphene. Further thermal annealing further increased the Dirac point (135.8 ± 22.7 V at 200 °C, > 180 V at 300 °C) (Supplementary Fig. 5); this change implies that the doping effect is increased by thermal annealing. Hole concentrations calculated using the shift in Dirac point also increased as $T_a$ increased (pristine graphene: $n = 4.46 \times 10^{11}\,cm^{-2}$, $T_a = 100\,°C$: $n = 7.65 \times 10^{12}\,cm^{-2}$, $T_a = 200\,°C$: $n = 9.78 \times 10^{12}\,cm^{-2}$, $T_a = 300\,°C$: $n > 1.296 \times 10^{13}\,cm^{-2}$) (Fig. 3e).[32]

These results also support that rearrangement of PFSA at high $T_a$ influences the $n$ of PFSA-doped graphene because the number of acidic protons in the proximity of graphene increases. All measurements showed that thermal annealing caused $R_{sh}$ decrease, WF increase and $n$ increase in PFSA-doped graphene as $T_a$ increased; these results are direct evidence that increased $T_a$ influences the doping effect of PFSA.

**Chemical and air stability**. To quantify the doping stability of PFSA under various conditions, we compared $R_{sh}$ changes of PFSA-doped and $HNO_3$-doped graphene under exposure to chemicals, and ambient conditions. To demonstrate the chemical invulnerability of PFSA-doped graphene, polar protic (deionized

water, isopropyl alcohol (IPA)), polar aprotic (dimethyl sulfoxide), and non-polar solvent (toluene) were spin-coated on p-doped graphene samples, which were then gently annealed to remove residual solvents. All treatments significantly increased the $R_{sh}$ of $HNO_3$-doped graphene on Si/SiO$_2$ substrate, but had negligible effect on $R_{sh}$ in the PFSA-doped graphene (Fig. 4a and Supplementary Table 2).

We also tested the chemical stability by dipping $HNO_3$-doped or PFSA-doped graphene into a solution of strong acid (hydrochloric acid, $K_a > 1$), weak acid (acetic acid, $K_a$: ~1.8 × 10$^{-5}$), weak base (ammonium hydroxide, $K_b$: ~1.8 × 10$^{-5}$), or strong base (sodium hydroxide, $K_b > 1$) for 15 s. $R_{sh}$ of $HNO_3$-doped graphene greatly increased after acid and base treatments (Fig. 4b and Supplementary Table 3). Especially, $R_{sh}$ of $HNO_3$-doped graphene after sodium hydroxide treatment was unmeasurable using the 4-point probe method; this result can be attributed to tearing of the surface of $HNO_3$-doped graphene by treatment with strong base chemicals (Supplementary Fig. 6a). In contrast, the surface of PFSA-doped graphene remained smooth even in strong base solution, so $R_{sh}$ of this graphene increased much less than it did in $HNO_3$-doped graphene (Supplementary Fig. 6b). These results demonstrate that p-type doping using macromolecular acid is stable against almost every chemical environment.

To verify the long-term air-stability of PFSA, we monitored the $R_{sh}$ change of PFSA-doped and $HNO_3$-doped graphene under ambient conditions (Fig. 4c). Although $HNO_3$ doping reduced the $R_{sh}$ of graphene to ~38.3% of the $R_{sh}$ of pristine graphene, $R_{sh}$ increased over 4 days to ~74.1% the $R_{sh}$ of pristine graphene. On the contrary, the PFSA-doped graphene showed decreased $R_{sh}$ to

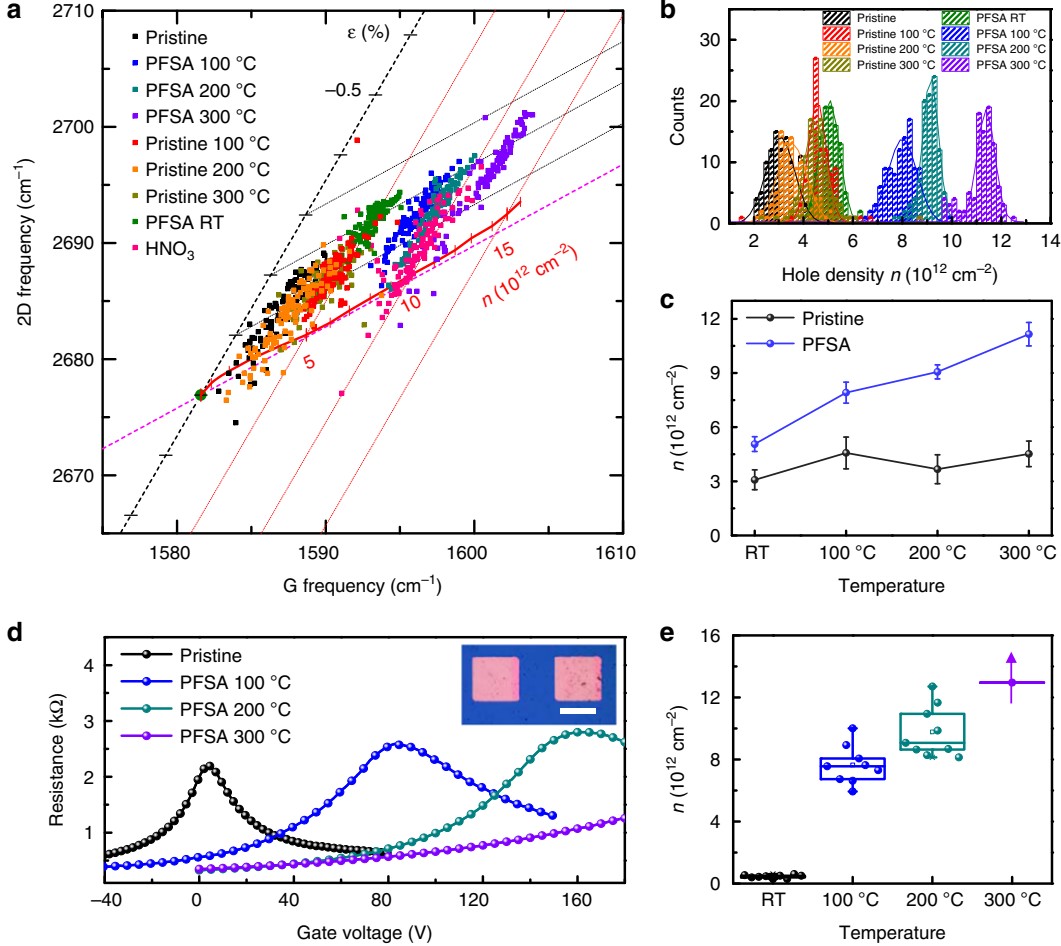

**Fig. 3** $n$ changes of PFSA-doped graphene induced by thermal annealing. **a** Variations in 2D- and G- bands of pristine, $HNO_3$-doped and PFSA-doped graphene. **b** Histogram of $n$, and **c** averaged $n$ of thermally annealed pristine and PFSA-doped graphene with various $T_a$ calculated from Raman spectroscopy results. The error bars represent the s.d. of multiple measurement results. **d** Current vs. voltage characteristics (inset: optical microscopy image of graphene-based FET; scale bar, 200 μm), and **e** $n$ of pristine, and PFSA-doped graphene annealed at various $T_a$ calculated from Dirac point shift. Each box chart includes minimum, lower quartile (lower horizontal line), median (middle horizontal line), mean (hollow square), upper quartile (upper horizontal line), maximum, and discrete data

~55.4% compared to the $R_{sh}$ of pristine graphene; this reduction in $R_{sh}$ was retained for 64 days without noticeable change under ambient conditions. Before the investigation of graphene's doping stability, pristine graphene samples were vacuum-annealed at 500 °C to eliminate inevitable p-type doping effects of residual dopants (e.g., PMMA, etchant, oxygen, water molecules)[1,51] which were introduced during transfer process. We measured contact angles by using deionized water and diiodomethane on pristine, $HNO_3$- and PFSA-doped 4LG to calculate surface energies of the graphene according to p-type doping (Fig. 4c and Supplementary Fig. 7); the Owens−Wendt model was used to calculate the surface energy[52,53]. The water contact angle of pristine 4LG was 55.3°; it increased to 96.7° when PFSA was doped on it; this change indicates that PFSA doping increased the hydrophobicity of the graphene surface. The calculated surface energy of the PFSA-doped graphene was ~21.69 mJ m$^{-2}$ (Supplementary Fig. 7 and 8). The hydrophobic surface with low surface energy of the PFSA-doped graphene can more stably maintain the p-type doping effect in ambient conditions than can $HNO_3$-doped 4LG (surface energy: 41.20 mJ m$^{-2}$). Similarly, increased WF by PFSA doping was also stably maintained for 10 days under ambient conditions (Fig. 4d), whereas the WF of

$HNO_3$-doped graphene decreased from 4.57 eV to 4.43 eV after exposure to ambient conditions for 8 days (Fig. 4e).

We also performed conductive atomic force microscopy (c-AFM) (Supplementary Fig. 9) to spatially resolve the electrical properties of graphene films according to doping methods. In-plane conductance of $HNO_3$-doped and PFSA-doped graphene were monitored after 5 days in ambient conditions by measuring current between p-doped graphene and Sb-doped Si tip. p-Type doping with $HNO_3$ or PFSA did not affect the surface topography of graphene. PFSA-doped graphene film showed uniform current in surface mapping, but the $HNO_3$-doped graphene surface showed significant heterogeneity of in-plane conductance (Supplementary Fig. 9b).

Calculated $n$ from Raman spectroscopy (Supplementary Fig. 10) and FET (Supplementary Fig. 11) results quantitatively prove that PFSA doping of graphene is stable under ambient conditions. The outstanding doping stability of PFSA on graphene can be attributed partly to the affluent fluorinated alkyls in PFSA, which substantially reduces the surface energy of the PFSA-doped graphene surface; it would repulse the molecules of applied solvents or ambient air, and maintain the doping effect[54].

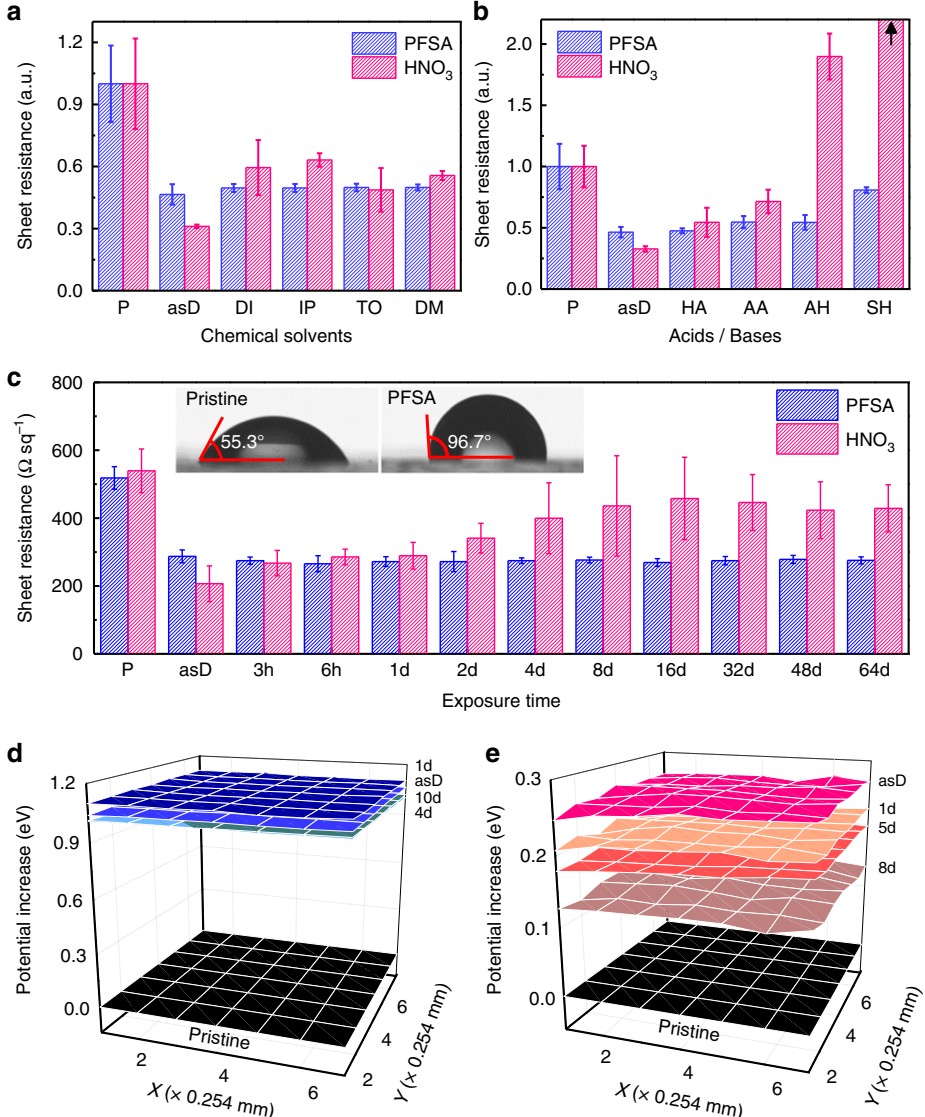

**Fig. 4** Chemical and ambient stability. $R_{sh}$ changes under **a** various solvent treatments (DI deionized water, IP isopropyl alcohol, DM dimethyl sulfoxide, TO toluene), **b** acid and base treatments (HA: hydrochloric acid; AA: acetic acid; AH: ammonium hydroxide; SH: sodium hydroxide), and **c** ambient conditions as a function of exposure time (inset: image of water droplet and contact angle on PFSA- and HNO$_3$-doped graphene). WF changes under the ambient exposure of **d** PFSA-doped graphene, and **e** HNO$_3$-doped graphene. The error bars represent the s.d. of multiple measurement results

The simplest PFSA-graphene configuration has binding energy of 0.79 eV, which is much larger than that of HNO$_3$-graphene (0.33 eV)[33]. The configurations between each dopant and graphene suggest that the acidic proton of HNO$_3$ is more exposed to outer circumstances than is the acidic proton of PFSA (Supplementary Fig. 4). In contrast to the parallel HNO$_3$-graphene configuration (Supplementary Fig. 4a, b), PFSA has non-planar molecular configuration (Supplementary Fig. 4d, e). The non-planar configuration and higher binding energy between PFSA and graphene improves doping stability over that of HNO$_3$ does; this coincides well with our experimental observations. Furthermore, these charge density calculations used the simplest molecule of PFSA, so binding energy between the dopant and graphene could be underestimated. Indeed, DFT calculation indicates that an increase in the length of the PFSA molecule gradually increases binding energy between the molecule and graphene (Supplementary Fig. 12, Supplementary Table 4, Supplementary Note 7). Therefore, the actual doping stability of macromolecular PFSA-doped graphene could be much higher than the calculation suggests.

X-ray photoelectron spectroscopy of p-doped graphene shows an intense F1s peak (~690 nm), and S2p peak (~170 eV) (Supplementary Fig. 13a); these peaks confirm that the PFSA molecule remains on the graphene surface: The C1s spectrum of PFSA-doped graphene revealed C−C $sp^2$ bonding (~284.7 eV), with four PFSA-related chemical bonds (i.e., C−O–C (~286.5 eV), C−S (~289.6 eV), −CF$_2$ (~292.4 eV), −CF$_3$ (~294.0 eV))[23,55]. which can be confirmed in the chemical structure of PFSA (Supplementary Fig. 13b, c).

Graphene doping using other macromolecules (PMMA and PEDOT:PSS) showed weak p-type doping effect and poor ambient stability, respectively (Supplementary Figs. 14 and 15); these results exhibit the uniqueness of the graphene doping using PFSA (Supplementary Note 9 and 10).

Considering all the experimental observations and further analyses of PFSA-doped graphene, we can conclude that PFSA meets all the requirements of an ideal p-type dopant to produce a graphene anode: (1) large $R_{sh}$ decrease, (2) substantial increase in surface WF, (3) high stability against high temperature, chemicals, and ambient conditions, (4) smooth and uniform

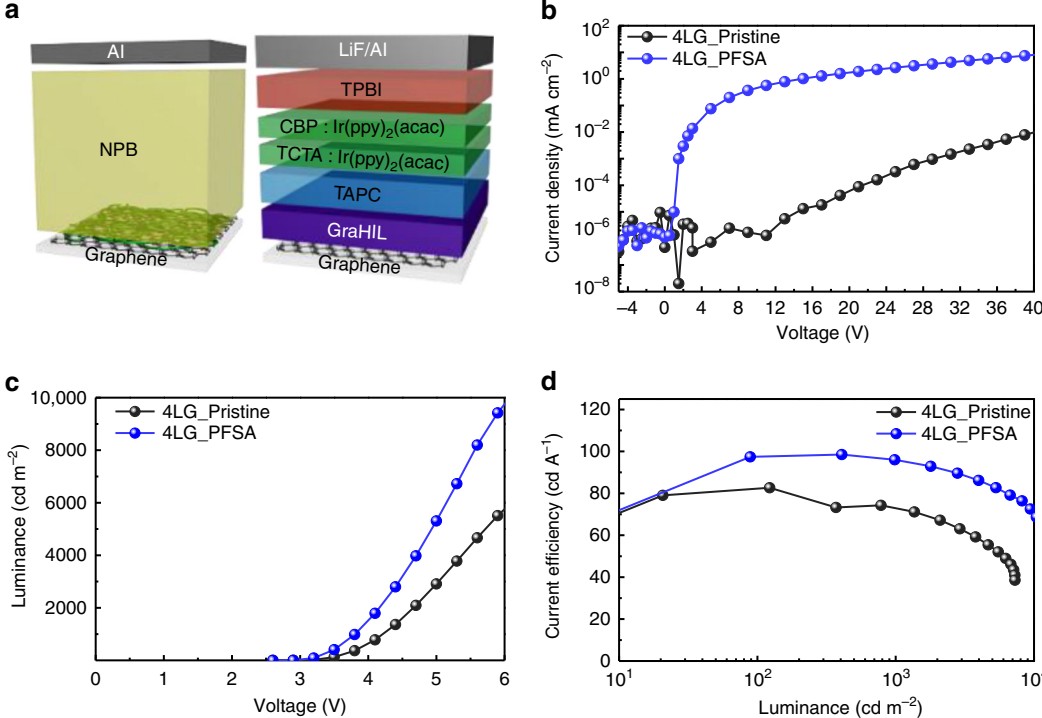

**Fig. 5** Optoelectronic application. **a** Schematic device structure of HOD and OLED. **b** Current density vs. voltage characteristics of HODs. **c** Luminance vs. voltage, and **d** current efficiency vs. luminance characteristics of green phosphorescent OLEDs with pristine and PFSA-doped graphene anode

surface without any defects or large particles, and (5) negligible decrease in OT.

**Optoelectronics application**. To demonstrate the improved hole-injection capability of the PFSA-doped graphene anode in organic optoelectronics, we fabricated hole-only devices (HODs) and green-emitting phosphorescent OLEDs that had anode made of pristine or PFSA-doped 4LG (Fig. 5a). Because of the large energy difference (> 1.0 eV) between the highest occupied molecular orbital energy level of the N,N'-Di(1-naphthyl)-N,N'-diphenyl-(1,1'-biphenyl)-4,4'-diamine (NPB) (~5.4 eV) and WF of pristine graphene (~4.3 eV), hole injection from the pristine graphene anode is seriously obstructed, so an HOD with the pristine 4LG showed very low current density. When the graphene was doped with PFSA, the hole current density in the HOD was dramatically increased by a factor of > $10^3$ due to a ~0.8 eV increase in the surface WF and reduced $R_{sh}$ (Fig. 5b). Hole injection efficiencies ($\eta$) were calculated (see Supplementary Note 11 for detail), and $\eta$ of the HOD with PFSA-doped graphene was > $10^3$ higher than in the HOD with pristine graphene (Supplementary Fig. 16). We uniformly spin-coated the polymeric hole-injection layers diluted with IPA on hydrophobic PFSA-doped graphene surface to fabricate green-emitting phosphorescent OLEDs (Supplementary Fig. 17). The OLED with the PFSA-doped graphene anode also had higher current density than did the OLED that had a pristine 4LG anode; this result was also caused by improved hole injection from graphene anode due to the increased surface WF of PFSA-doped graphene (Supplementary Fig. 18a). The OLED with PFSA-doped 4LG had lower operating voltage than the OLED with pristine 4LG, because PFSA-doped graphene anode has lower $R_{sh}$ and higher hole injection capability than did pristine graphene anode (Fig. 5c). As a result, the device with PFSA-doped 4LG also showed higher current efficiency (CE ~98.5 cd A$^{-1}$) and higher power efficiency (PE ~95.6 lm W$^{-1}$) without an out-coupling structure than did the device with the pristine 4LG

(~82.7 cd A$^{-1}$ and ~77.6 lm W$^{-1}$) (Fig. 5d, Supplementary Fig. 18b). The improved electroluminescent properties of OLED with the PFSA-doped graphene demonstrate the possibility of using PFSA-doped graphene as flexible anode to simultaneously reduce operating voltage and increase luminous efficiency.

## Discussion

We used a macromolecular fluorinated acid, PFSA, as a chemical p-type dopant to give extremely stable chemical p-type doping for graphene. The PFSA-doped graphene met the requirements for ideal p-type doping of graphene anode: (1) large $R_{sh}$ decrease, (2) substantial increase in surface WF, (3) high stability against all kinds of circumstances (high temperatures, chemicals, and ambient conditions), (4) smooth and uniform surface, and (5) negligible decrease in OT. The non-volatility, strong binding to graphene, and chemical and thermal stability of PFSA can explain the excellent environmental stability of PFSA-doped graphene.

We also fabricated HODs and OLEDs to demonstrate the superior hole injection and electroluminescent characteristics of devices that used the PFSA-doped graphene as an anode. An HOD that used the PFSA-doped graphene showed dramatic improvement of hole current, and OLEDs that used PFSA-doped graphene exhibited increase of luminous efficiencies: this result demonstrates the possibility of practical anode application of the PFSA-doped graphene due to its improved electrical conductivity and surface WF, and confirms that our PFSA is a promising p-type chemical dopant to make more ideal flexible graphene electrodes with excellent environmental stability, high WF; simultaneous achievement of these attributes has been almost impossible using conventional doping with small molecules.

This work provides a promising way to overcome the demerits of chemically doped graphene electrodes and is a significant step towards development of stable graphene electrodes for practical use in various opto-electronic devices.

## Methods

**Fabrication and characterization of graphene anodes**. SLG was synthesized on Cu foil by CVD. The foils were heated to 1060 °C with 15-sccm flow of $H_2$ gas and annealed for 30 min. As a carbon source, 60 sccm of $CH_4$ gas was flowed for 30 min, then the Cu foil was rapidly cooled to room temperature. After synthesis of graphene on Cu foil, PMMA was applied by spin-coating polymer solution (996k, purchased from Sigma Aldrich) as a supporting polymer layer for graphene transfer. $O_2$-plasma treatment using reactive ion etching (RIE) was performed to remove the graphene that grew on the bottom of the Cu foil. The foil was immersed in a CE100 (YMS tech) etchant solution for 1 h to etch the Cu foil, then rinsed with deionized water for several hours to remove the etchant residue. Floated SLG was transferred onto the target substrate. Before measuring the stability, we performed vacuum annealing on transferred graphene at ~500 °C. Glass substrates were used to investigate the OT, large-area uniformity, and ambient stability. To perform c-AFM, we used native Si as substrates. In other characterizations of PFSA doping effect (e.g., Raman spectroscopy, chemical-/thermal- stability test), we used $Si/SiO_2$ (~300 nm) substrate. The supporting polymer layer was removed by soaking in an acetone bath. 4LG was stacked by repeating this process. Then 0.1 wt.% tetra-fluoroethylene-perfluoro-3,6-dioxa-4-methyl-7octenesulfonic acid copolymer (CAS number: 31175-20-9, Sigma-Aldrich) in IPA was spin-cast to chemically dope the graphene, then the sample was annealed at 100 °C for 10 min to remove the solvent. Raman spectra were obtained using a home-built setup operated with a 514-nm laser[50]. $R_{sh}$ of graphene electrodes was measured using a 4-point probe combined with a Keithley 2400 source meter. $R_{sh}$ decrease of PFSA-doped graphene saturates at PFSA thickness of ~10 nm, and we used 3.4 nm-thick PFSA layer for chemical doping of graphene. Surface potential difference was measured using an SKP-5050 Kelvin Probe measurement system. OT was measured using a SCINCO S-3100 UV-Vis spectrophotometer. To probe the morphology, conducting atomic force microscopy (c-AFM) was measured using a Bruker Dimension Icon Scanning Probe Microscope equipped with TUNA module in contact mode with a 0.01 −0.025 Ω·cm Antimony (n) doped Si tip (SCM-PIC, Bruker). A 300-nm-thick layer of $SiO_2$ on p-doped Si was used as substrate for graphene-based FETs. Pristine graphene was transferred onto $SiO_2$ substrate, then Cr (5 nm)/Au (300 nm) layers were thermally deposited using a pre-patterned mask to form FETs of width 200 μm and length 400 μm. XPS and UPS measurement of pristine and PFSA-doped graphene samples were conducted using the same equipment in collaboration with the Korea Basic Science Institute.

**Fabrication of green phosphorescent OLED devices**. To prepare graphene as a transparent conducting electrode in OLEDs, we formed a 4LG anode by stacking SLGs. The prepared multi-layered graphene was patterned by RIE using $O_2$ plasma and a pre-patterned shadow mask. UV-ozone treatment was performed for 10 min to activate the graphene films and to uniformly deposit the GraHIL on the graphene anode. The polymeric hole-injection layer was composed of Poly(3,4-ethylenedioxythiophene):poly(styrenesulfonate) (PEDOT:PSS) (CLEVIOS P VP AI4083) and PFSA, and was spin-coated to give a 50-nm-thick layer on the graphene anode, then annealed on a hot plate at 150 °C for 30 min. A 15-nm-thick layer of Di-[4-(N,N-ditolyl-amino)-phenyl]cyclohexane (TAPC) was deposited on top of the HIL as a hole transporting material. A 5-nm-thick layer of 1,1-bis[4-5 [N,N-di(p-tolyl)amino]phenyl]cyclohexane (TCTA) doped with green phosphorescent dopants (bis(2-phenylpyridine)iridium (III) acetylacetonate ($Ir(ppy)_2(acac)$) (97:3 by volume) and a 5-nm thick layer of 4,4′-N,N′-dicarbazolylbiphenyl (CBP) doped with $Ir(ppy)_2(acac)$ (96:4 by volume)) were deposited successively as a green-emitting layer. Then a 55-nm-thick layer of 1,3,5-tri(phenyl-2-benzimidazolyl)-benzene (TPBI) was deposited as an electron-transporting layer. Finally, LiF (1 nm)/Al (100 nm) was deposited on top of the TPBI layer as a cathode. The devices were encapsulated using a glass lid and epoxy resin. A Keithley 236 source measurement unit and Minolta CS 2000 spectroradiometer were used to measure current-voltage-luminance characteristics of green phosphorescent OLEDs.

**Calculation methods**. All DFT calculations were performed using projector augmented wave pseudopotentials with the Perdew−Burke−Ernzerhof type of generalized gradient approximation functions, which is implanted in the Vienna Ab-initio Simulation Package[56]. The experimental system was modeled as one PFSA molecule doped on each 4 × 4 graphene layer; this doping density is the most appropriate for the system. To avoid the undesirable interaction from the periodic supercell images, we used a supercell as large as 9.84 Å × 9.84 Å × 35 Å within potential dipole corrections for all calculations. The plane-wave energy cutoff was 650 eV, and the Brillouin-zone sampling meshes were 7 × 7 × 1. For the geometry optimization, all the coordinates were fully relaxed until the forces were < 0.01 eV Å$^{-1}$. The Tkatchenko−Scheffler Van der Waals correction[57] was used to describe the dispersion interaction between the adsorbate molecule and graphene layer.

The adsorption energy per PFSA molecule is defined as:

$$E_{ad} = E_{PFSA\text{-}graphene} - E_{PFSA} - E_{graphene},$$

where $E_{PFSA\text{-}graphene}$ is the total energy of PFSA-doped graphene, $E_{PFSA}$ is the total energy of isolated PFSA molecule, and $E_{graphene}$ is the total energy of the pristine graphene layer.

**Data availability**. The datasets generated during the current study are available from the corresponding authors upon reasonable request.

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

## Acknowledgements

This work was supported by the National Research Foundation of Korea (NRF) grant funded by the Korea government (Ministry of Science, ICT & Future Planning) (NRF-2016R1A3B1908431), by the Nano Material Technology Development Program through the National Research Foundation of Korea (NRF) funded by the Ministry of Science, ICT & Future Planning (MSIP, Korea) (NRF-2014M3A7B4051747) and by National Honor Scientist Program (2010-0020414). This work was also supported by 10079974 Development of core technologies on materials, devices, and processes for TFT backplane and light emitting frontplane with enhanced stretchability above 20%, with application to stretchable display, and LG Display under LGD-SNU Incubation Program.

## Author contributions

S.-J.K. and T.-H.H. designed the study, performed experiments, analyzed data, and prepared the manuscript. T.Y.K. and S.R. performed and analyzed Raman spectroscopy. N.L. and K.S.K. performed density functional theory calculation. Y.K., D.J.K. and B.H.H. fabricated field effect transistors and analyzed the results. S.-H.B. and Y.Y. performed and analyzed conductive AFM. T.-W.L. designed and supervised the study, analyzed the data and prepared the manuscript. All authors discussed the results and commented on the manuscript.

## Additional information

**Competing interests:** The authors declare no competing interests.

