## [Peer Review File · Nature Communications]

Reviewers' comments:

Reviewer #1 (Remarks to the Author):

In the manuscript "Extremely stable graphene electrodes doped with macromolecular acid" Kwon et al. introduces a new method for doping graphene using a macromolecular acid. While the approach has certain advantages, such as a smoothing effect on the graphene surface, it is interesting to a very limited group of researchers and should be published in a more specialized journal. The reason for my assessment is the limited improvement over existing doping techniques. The increase in carrier concentration to 10^{13} is routinely achieved by metal-chloride doping (see DOI: 10.1039/c6ra04449b) as is a workfunction change of 0.7-0.8 (see doi:10.1038/s41598-017-09465-x). Long-term stability can be achieved with small molecules (see doi:10.1088/0957-4484/25/39/395701).

Moreover, the two central claims of the paper, long-term stability and large increase of carrier concentration, are not well supported by the experimental methods and should be improved: First, Hall-effect measurements and FET measurements should be carried out to extract carrier concentration and direct work function measurements using Kelvin probe measurements should be carried out. Second, a more substantiated reason for the claimed enhanced environmental stability of large molecules should be provided since experience from graphene n-type doping suggests that a low volatility is not sufficient for stable doping.

To keep the focus of the paper distracting parts, such as XPS, LED, absence of strain, and DFT should be moved to the supplementary information.

Reviewer #2 (Remarks to the Author):

The authors report an extremely environmentally-stable graphene electrode doped with macromolecular acid PFSA as a chemical p-type dopant, resulting in a stretchable, conductive, and transparent materials. The chemically doped graphene to achieve high conductivity have been developed for the decade. However, most works are not stable, thus this work shows high impact in this research field.

Overall, the manuscript was well organized and written. Also, the study was well conducted and the discussion is convincing. This referee recommends to accept and publish the article after clarifying some arguments as shown below.

1. There is a similar chemical dopant TFSA, which has similar p-type doping on graphene. Although it is a small molecular, the typical papers should be well cited, such as Nano Lett., 2012, 12 (6), pp 2745–2750, Nanotechnology. 2011 Oct 21;22(42):425701, RSC Adv., 2016, 6, 32746
2. In page 4, line 81: ...Since the development of polymeric acid (i.e., poly(styrene81 sulfonic acid)) doping of conducting polymers greatly improved its processability and stability. What're the features when compare the performance and stability of this work with these reported graphene-PSS:PEDOT composites?
3. Page 6 Line 121, weaken the statement "without defect" to "lower defect". There is no evidence to show this graphene is single crystalline.
4. The statement: "... this result proves that solution-processed doping of the PFSA does not induce any significant structural defects in the graphene lattice...". The authors should provide C1s, S and F1s spectra from XPS to he detail chemical bonding states of C-F or C-S.
5. The authors used normalized Rsh to define the doping performance over the whole manuscript. However, the value of Rsh of their pristine 1L and 4L graphene should be presented before this normalization. Also, the author find a good approach (c-AFM) to evaluate the uniformity of the doped graphene. However, the AFM is limited due to their quite localized characterization, when considering the large area uniformity, this may not provide additional information. The authors could provide the spatial mapping of Rsh to demonstrate this factor they could provide OLED

lighting image to direct evidence this electrical uniformity.

6. Figure 2 a and b require error bar just like Fig2c.

7. Fig 2 c, the PFSA doped graphene shows the hydrophobic property, which was due to the fluorinated groups. The might be another factor to repulse the foreign molecular and then enhance the chemical stability.

8. The authors claim the chemical stability of doped graphene, while the experiential test only organic solvents. Can this dopant sustain well to acid or base chemicals?

9. Line 198, correct to "10um²x10um²."

10. Line 201, the concentration(n) of doped graphene is~ 2 times higher than that of pristine graphene. Normally it should be 1 or 2 order increase to achieve significant WF shifting and Rsh reduction. Can it be well explained? Will it be better to employ the hall measurement to probe this N? The Raman single may also effect by the underlying substrate(localized strain).

11. Line 210, the statement" this also indicates that sulfonic acid groups (-SO₃H) would be rearranged towards underlying graphene, and that thereby cause increase in n of PFSA-doped graphene." The authors provide a good DFT calculation to evidence this argument. However, this argument can be directly evidenced by c-AFM to understanding this potential difference before or after 300oC treatment.

12. Line 287. Since the doped graphene shows highly hydrophobic surface. How the stacking film can be uniform formed by coating hydrophilic PEDOT:PSS on doped graphene?

Reviewer #3 (Remarks to the Author):

Authors investigated the effect of PFSA doping on graphene. The interesting points are high p-type doping stability under ambient atmosphere and at elevated temperature up to 300 oC. The sheet resistance of p-doped graphene decreased as a function of annealing temperature, which is strikingly different with typical small molecule doping. By changing work function caused by p-type doping, the efficiency of OLEDs are much improved. Therefore, I recommend that this paper will be published in Nature Commun. after minor revision.

Comments:

1. The substrate for this study is not easy to be recognized. What kind of substrate is used for doping study?

2. Figure quality should be improved. For instance, The labels in x axis do not look good in Figure 3c and d

Response to Reviewers' comments

We appreciate the reviewers' valuable comments. We revised our manuscript to comply with all of them, and have prepared point-by-point responses, which we present here. The revised parts in the manuscript are marked in red.

Reviewer #1

Comments:

In the manuscript “Extremely stable graphene electrodes doped with macromolecular acid” Kwon et al. introduces a new method for doping graphene using a macromolecular acid. While the approach has certain advantages, such as a smoothing effect on the graphene surface, it is interesting to a very limited group of researchers and should be published in a more specialized journal. The reason for my assessment is the limited improvement over existing doping techniques. The increase in carrier concentration to 10^{13} is routinely achieved by metal-chloride doping (see DOI: 10.1039/c6ra04449b) as is a workfunction change of 0.7-0.8 (see doi:10.1038/s41598-017-09465-x). Long-term stability can be achieved with small molecules (see doi:10.1088/0957-4484/25/39/395701). Moreover, the two central claims of the paper, long-term stability and large increase of carrier concentration, are not well supported by the experimental methods and should be improved: First, Hall-effect measurements and FET measurements should be carried out to extract carrier concentration and direct work function measurements using Kelvin probe measurements should be carried out. Second, a more substantiated reason for the claimed enhanced environmental stability of large molecules should be provided since experience from

graphene n-type doping suggests that a low volatility is not sufficient for stable doping. To keep the focus of the paper distracting parts, such as XPS, LED, absence of strain, and DFT should be moved to the supplementary information.

Answer) We thank the reviewer for the helpful comments. Due to the reviewer's comments, we now have a chance to solidify our motivation of research and clarify why our work is important and different from the previous works. We also added more sophisticated and substantiated data to support our main claim on enhanced doping stability. We also revised our manuscript to focus on the doping stability issue to comply with the reviewer's comment.

The reviewer mentioned other existing doping techniques [*RSC Adv.*, 2016, 6, 32476; *Sci. Rep.* 2017, 7, 9052; *Nanotechnology* 2014, 25, 395701] based on small molecules (e.g. AuCl₃) that achieve increase of carrier concentration and work function. By using AuCl₃ as a p-type dopant of graphene, the authors reported that the sheet resistance (R_{sh}) can be reduced to ~40 Ω/sq, and work function (WF) can be tunable in a wide range (~1.5 eV). Reported hole concentration of AuCl₃-doped single-layered graphene was $\sim 4 \times 10^{13} \text{ cm}^{-2}$ [*RSC Adv.* 2016, 6, 32476]. Although reported AuCl₃ doping showed substantial improvement in electrical properties, it has critical disadvantages (i.e., ambient instability, formation of protruding particles) that hinder its practical use in thin-film optoelectronic devices. Conventional p-type graphene dopants (e.g., AuCl₃, HNO₃, FeCl₃, TFSA.) showed unstable doping characteristics under ambient conditions. Among these conventional dopants, TFSA showed the best ambient stability compared with the other dopants, however, it still exhibited limited stability (~20% increase of R_{sh} in 400 h) [*Nanotechnology* 2014, 25, 395701]. Especially, long-term ambient stability results of AuCl₃-doped graphene showed considerable loss of conductivity (R_{sh} increased by a factor of ~4 in 60 d), which is unacceptable for

practical electrode application [*RSC Adv.* 2016, 6, 32476]. To improve the intrinsic instability of conventional doping methods using small-molecules, PMMA over-coating methods have been attempted [*Nanotechnology* 2014, 25, 395701]. However, the thickness of covered insulator film (*i.e.*, PMMA) on the p-doped graphene was $\sim 1 \mu\text{m}$, which significantly limits the use of this configuration as an anode in thin-film optoelectronics. Moreover, this attempt needs additional process to make p-type doped graphene stable; these processes complicate the overall doping process, and the ambient stability of the resultant graphene remains limited. Also, conventional small-molecule dopants are highly vulnerable to various solvents (*e.g.*, water, acetone, anisole, isopropanol, nitromethane). Among the small molecular p-type dopants that have been tested, FeCl_3 showed the best chemical stability against the solvents [*Nanotechnology* 2014, 25, 395701]. However, FeCl_3 -doped graphene showed substantial increase ($\sim 150\%$) of R_{sh} after immersion in water; this result means that FeCl_3 doping still has limited chemical stability. Moreover, thermal treatment on AuCl_3 doping accelerates the degradation of the p-type doping effect [*RSC Adv.* 2016, 6, 32476]. Thermal treatment on AuCl_3 -doped graphene to $80 \text{ }^\circ\text{C}$, and $140 \text{ }^\circ\text{C}$ substantially increase the R_{sh} of the doped graphene after 3 h ($76.2 \text{ } \Omega/\text{sq}$ to $255.9 \text{ } \Omega/\text{sq}$ at $80 \text{ }^\circ\text{C}$, and $316.8 \text{ } \Omega/\text{sq}$ at $140 \text{ }^\circ\text{C}$). Therefore, our chemical graphene doping using PFSA is the first to maintain the doping effect for long time (about 2 months), and has the advantage that the doping effect is intensified by thermal treatment. To our best knowledge, no previous report meet almost every aspect of stability (*i.e.*, ambient-/ thermal-/ chemical-) at the same time.

Chemical doping with metal chlorides includes the reduction of metal ions, and this process forms metal particles on the graphene surface [*Nat. Photon.* 2012, 6, 110; *RSC Adv.* 2016, 6, 32476]. These particles are more than several tens of nanometers tall, so they induce severe leakage current when metal chloride-doped graphene is applied in thin-film optoelectronics. For the ideal direction of graphene doping to be used in anode of opto-

electronics, outstanding doping stability, and surface uniformity is essentially needed as well as improvement of electrical and electronic properties (*i.e.*, R_{sh} , WF). Our work using PFSA as a chemical doping for graphene not only showed improvement of electrical and electronic properties (~56% reduction of R_{sh} , ~0.8 eV increase of WF), but surface flatness (root mean square roughness: 0.495 nm), and extreme stability against all kinds of aspects (*i.e.*, ambient-/thermal-/chemical- challenges).

To support our central claims (*i.e.*, long-term stability, large increase of carrier concentration), We estimated hole concentration from electrical characteristics of pristine and PFSA-doped graphene field-effect transistors (FETs). The substrate was 300 nm-thick SiO₂ on p-doped Si. Pristine graphene were transferred onto SiO₂ substrate, then Cr (5 nm) and Au (300 nm) were thermally deposited using a pre-patterned mask to form FETs of width 200 μm and length 400 μm (Figure R1a, inset). Our FETs using pristine graphene had a Dirac point of 6.2 ± 1.5 V. PFSA layers were directly spin-cast on FETs and thermally annealed to p-type dope the pristine graphene (Figure R1b). Under thermal annealing at annealing temperature (T_a) = 100 °C, Dirac point substantially increased to 106.2 ± 17.2 V; this change indicates that PFSA strongly p-doped the graphene. Further thermal annealing further increased the Dirac point (135.8 ± 22.7 V at $T_a = 200$ °C, and > 180 V at $T_a = 300$ °C); this continuing change implies that thermal annealing increases the doping effect. Hole concentration (\mathbf{n}) was calculated from the Dirac point results as

$$\mathbf{n} = -\alpha(V_g - V_{CNP}),$$

where $\alpha = 7.2 \times 10^{11} \text{ cm}^2/\text{V}$ [ACS Nano 2014, 8, 868-874] and V_{CNP} is the gate voltage at which the charge neutrality point appear. The result was $\mathbf{n} = 4.46 \times 10^{11} \text{ cm}^{-2}$ (pristine graphene), $7.65 \times 10^{12} \text{ cm}^{-2}$ (PFSA-doped, $T_a = 100$ °C), $9.78 \times 10^{12} \text{ cm}^{-2}$ (PFSA-doped, $T_a = 200$ °C), and $> 1.296 \times 10^{13} \text{ cm}^{-2}$ (PFSA-doped, $T_a = 300$ °C) (Figure R1c). Ambient stability

of PFSA doping was monitored by measuring the electrical characteristics of the FETs with PFSA-doped graphene (Figure R1d). The upshift of the Dirac point of PFSA-doped graphene did not decrease, and even slightly increased after the sample was held in ambient condition for 21 d (Figure R1e). Calculated average hole concentration (PFSA as-doped: $7.65 \times 10^{12} \text{ cm}^{-2}$, 7 d: $9.14 \times 10^{12} \text{ cm}^{-2}$, 21 d: $1.03 \times 10^{13} \text{ cm}^{-2}$) also prove that PFSA doping of graphene is stable in ambient condition (Figure R1f).

Figure R1. Hole concentration changes of PFSA-doped graphene calculated by Electrical characteristics of FETs. (a) Current versus voltage characteristics (inset: optical microscopy image of graphene-based FET (scale bar, 200 μm)), (b) statistics of Dirac voltage, and (c) hole concentrations of pristine graphene (black), PFSA-doped graphene and 100 $^{\circ}\text{C}$ annealed (red), 200 $^{\circ}\text{C}$ annealed (green), and 300 $^{\circ}\text{C}$ annealed (blue). (d) Current versus voltage characteristics, (e) statistics of Dirac voltage, and (f) hole concentrations of as-doped (red) and after ambient exposure for 7 d (purple), and 21 d (olive).

To measure the WF directly, we performed Kelvin probe measurement using SKP-5050 Kelvin probe system (Figure R2a). We performed potential difference mapping of pristine and PFSA-doped graphene. Pristine, and PFSA-doped graphene showed uniform WF over a large area ($> 1.5 \text{ mm} \times 1.5 \text{ mm}$), and determined that that PFSA doping is uniform over the large area. PFSA-coating and annealing at $T_a = 100 \text{ }^{\circ}\text{C}$ yielded $\sim 0.73 \text{ eV}$ increase of surface potential, which is consistent with ultraviolet photoelectron spectroscopy (UPS) results (figure 4c). Further thermal annealing (*i.e.*, $T_a = 200 \text{ }^{\circ}\text{C}$, $T_a = 300 \text{ }^{\circ}\text{C}$) caused additional potential increase: $\sim 0.82 \text{ eV}$, and $\sim 1.01 \text{ eV}$ for 200 $^{\circ}\text{C}$, and 300 $^{\circ}\text{C}$, respectively. Considering the UPS results, WF of pristine was $\sim 4.3 \text{ eV}$, and those of PFSA were 5.03 eV at $T_a = 100 \text{ }^{\circ}\text{C}$, 5.12 eV at $T_a = 200 \text{ }^{\circ}\text{C}$, and 5.31 eV at $T_a = 300 \text{ }^{\circ}\text{C}$. Annealed PFSA doped graphene was stored in ambient condition and WF was measured after 4, 6, and 9 days using the same measurement tools. Increased WF (5.31 eV) obtained by PFSA doping $T_a = 300 \text{ }^{\circ}\text{C}$ showed insignificant change even after 10 d. This result also indicates that doping effect of PFSA on graphene is stably maintained in ambient conditions.

Figure R2. WF of p-type doped graphene. (a) Surface potential mapping of pristine graphene (black), PFSA-doped and annealed at 100 °C (red), 200 °C (green), or 300 °C (blue), and (b) pristine graphene (black), PFSA-doped and 300 °C annealed (blue), (b) WF changes of PFSA-doped graphene at various annealing temperature. (c) Surface potential mapping of pristine graphene, as HNO₃-doped graphene, and changes after exposure in

ambient. (d) *WF* changes of HNO₃-doped graphene after ambient exposure. (e) Surface potential mapping of pristine graphene, as-PFSA-doped graphene, and changes after exposure in ambient. (f) *WF* changes of PFSA-doped graphene after exposure to ambient conditions.

We propose a more-substantiated reason for the exceptional stability of PFSA doping on graphene. PFSA is macromolecule, so it is non-volatile, which directly indicates that PFSA is stable at the thermal energy that is available at room temperature. Binding energy calculation from DFT calculation (Table R1) confirms that increase in molecular size increases binding energy with graphene.

Compound	Binding energy
HNO ₃	0.33
(CF ₃) ₂ CF-O-CF ₂ -(CF ₃)CF-O-CF ₂ CF ₂ SO ₃ H	0.79
CF ₃ -(CF ₂)-CF ₃ CF-O-CF ₂ -(CF ₃)CF-O-CF ₂ CF ₂ SO ₃ H	0.95
CF ₃ -(CF ₂ -CF ₂)-CF ₃ CF-O-CF ₂ -(CF ₃)CF-O-CF ₂ CF ₂ SO ₃ H	0.99

Table R1. Calculated binding energy [eV] of p-type dopants and graphene

Furthermore, PFSA contains many fluorinated alkyls (Figure R3a), and therefore has both hydrophobic and oleophobic properties, because a high concentration of fluorinated alkyl significantly reduces the surface energy [*Proc. Natl. Acad. Sci.* 2008, 105, 18200; *Thin Solid*

Films, 1993, 230, 209-216]. A water droplet had high contact angle ($> 90^\circ$) on PFSA-doped graphene (Figure R3b). Because of this combination of almost perfect non-volatility with low surface energy, PFSA doping on graphene has outstanding stability.

Figure R3. (a) Chemical structure of PFSA, and (b) Water contact angle of PFSA-doped graphene

DFT calculation provides us doping mechanism of our PFSA doping on graphene (*i.e.*, electronic dipole interaction of acidic proton in sulfonic acid with graphene); these results provide significant originality to our work, so we put some of these results in main manuscript, not in supplementary information. Alternatively, we moved further detailed DFT calculation using longer PFSA molecules, the results of OLEDs, and XPS data to the supplementary information as the reviewer commented.

Revised parts in the manuscript)

Page 10 in the manuscript,

“To quantify the p-type doping effect in PFSA-doped graphene, we also calculated the charge concentration (n) by using in-depth Raman spectrum analysis and the Dirac point shift in field-effect transistors (FETs).”

Page 10 in the manuscript,

“Notably, PFSA-doped graphene treated at $T_a = 300$ °C showed $n \sim 11 \times 10^{12}$ cm⁻², which is three times higher than n in a pristine graphene (Figure 3b,c). PFSA-doped graphene showed slight increase of strain (Pristine graphene: 0.1% to -0.2%, PFSA-doped graphene at $T_a = 300$ °C: 0.1% to -0.3%); this change can be attributed to slight thermal deformation during thermal annealing.

We also fabricated graphene-based FETs on Si/SiO₂ (300 nm) substrate (Figure S5a). FETs that used pristine graphene had a Dirac point of 6.2 ± 1.5 V. After PFSA doping and sequential thermal annealing, the Dirac point increased substantially to 106.2 ± 17.2 V; this change indicates that PFSA strongly p-doped the graphene. Further thermal annealing induced additional increase in the Dirac point (135.8 ± 22.7 V at 200 °C, > 180 V at 300 °C); this change implies that the doping effect is increased by thermal annealing. Hole concentrations calculated using the Dirac point shift also increased continuously with T_a (pristine graphene: $n = 4.46 \times 10^{11}$ cm⁻², $T_a = 100$ °C: $n = 7.65 \times 10^{12}$ cm⁻², $T_a = 200$ °C: $n = 9.78 \times 10^{12}$ cm⁻², $T_a = 300$ °C: $n > 1.296 \times 10^{13}$ cm⁻²).³²”

Page 11 in the manuscript,

“We calculated n again after 15 d from Raman spectroscopy results. n of PFSA-doped graphene did not change, but n of HNO₃-doped graphene decreased drastically (Figure 3d).

Electrical characteristics of the FETs with PFSA-doped graphene were measured again after exposure to ambient conditions (Figure S5b). The upshifted Dirac point of the PFSA-doped graphene did not decrease, but even slightly increased during exposure to ambient conditions for 21 d. Calculated n from Raman spectroscopy and FET results also prove that PFSA doping of graphene is stable under ambient conditions.”

Page 12 in the manuscript,

“DFT calculation also suggested that PFSA doping increases the WF of graphene by ~ 0.71 eV (Figure 4b), which is consistent with results of ultraviolet photoelectron spectroscopy (UPS) (~ 0.8 eV) (Figure 4c). We performed Kelvin probe measurement (Figure 4d,S9). Pristine, and PFSA-doped graphene showed uniform WF throughout the large area (> 1.5 mm \times 1.5 mm); *i.e.*, PFSA doping was uniform over the large area (Figure 4d). PFSA-coating and 100 °C annealing exhibited ~ 0.73 eV increase of surface potential, which is consistent with results of UPS and of DFT calculation. Thermal annealing increased the potential to ~ 0.82 eV with $T_a = 200$ °C, and to ~ 1.01 eV at $T_a = 300$ °C. Considering the UPS results, WF of pristine graphene was ~ 4.3 eV, and those of PFSA-doped graphene were 5.03 eV at $T_a = 100$ °C, 5.12 eV at $T_a = 200$ °C, and 5.31 eV at $T_a = 300$ °C. Increased WF by PFSA doping was maintained for 10 d in ambient air (Figure S9a,b), whereas the WF of HNO₃-doped graphene decreased from 4.57 eV to 4.43 eV after exposure to ambient air for 8 d (Figure S9c,d). This result also indicates that the doping effect of PFSA on graphene is stable in ambient conditions.”

Page 17 in the manuscript,

“A 300-nm-thick layer of SiO₂ on p-doped Si was used as substrate for graphene-based FETs. Pristine graphene was transferred onto SiO₂ substrate and Cr (5 nm)/ Au (300 nm) were thermally deposited using a pre-patterned mask to form FETs of width 200 μm and length 400 μm.”

Page 6 in the supporting information,

“We numerically calculated hole concentration changes by fabricating field-effect transistors (FETs) that use PFSA-doped graphene. The substrate was 300 nm-thick SiO₂ on p-doped Si. Pristine graphene were transferred onto SiO₂ substrate, then Cr (5 nm) and Au (300 nm) were thermally deposited using a pre-patterned mask to form FETs of width 200 μm and length 400 μm (Figure S5a, inset). Our FETs using pristine graphene had a Dirac point of 6.2 ± 1.5 V. PFSA layers were directly spin-cast on FETs and thermally annealed to p-type dope the pristine graphene (Figure S5b). Under thermal annealing at annealing temperature $T_a = 100$ °C, Dirac point substantially increased to 106.2 ± 17.2 V; this change indicates that PFSA strongly p-doped the graphene. Further thermal annealing further increased the Dirac point (135.8 ± 22.7 V at $T_a = 200$ °C, and > 180 V at $T_a = 300$ °C); this continuing change implies that thermal annealing increases the doping effect. Hole concentration (n) was calculated from the Dirac point results as

$$\mathbf{n} = -\alpha(V_g - V_{CNP}),$$

where $\alpha = 7.2 \times 10^{11} \text{ cm}^2/\text{V}^{\text{S1}}$ and V_{CNP} is the gate voltage at which the charge neutrality point appear. The result was $\mathbf{n} = 4.46 \times 10^{11} \text{ cm}^{-2}$ (pristine graphene), $7.65 \times 10^{12} \text{ cm}^{-2}$ (PFSA-doped, $T_a = 100$ °C), $9.78 \times 10^{12} \text{ cm}^{-2}$ (PFSA-doped, $T_a = 200$ °C), and $> 1.296 \times 10^{13} \text{ cm}^{-2}$ (PFSA-doped, $T_a = 300$ °C) (Figure S5c). Ambient stability of PFSA doping was monitored by measuring the electrical characteristics of the FETs with PFSA-doped graphene

(Figure S5d,e). The upshift of the Dirac point of PFSA-doped graphene did not decrease, and even slightly increased after the sample was held in ambient condition for 21 d. Calculated average hole concentration (PFSA as-doped: $7.65 \times 10^{12} \text{ cm}^{-2}$, 7 d: $9.14 \times 10^{12} \text{ cm}^{-2}$, 21 d: $1.03 \times 10^{13} \text{ cm}^{-2}$) also prove that PFSA doping of graphene is stable in ambient condition (Figure S5f).”

Page 14 in the manuscript, we moved the results of HODs and OLEDs from main manuscript to supplementary information as the reviewer suggested.

“To demonstrate the improved hole injection capability of the PFSA-doped graphene anode in organic optoelectronics, we fabricated hole-only devices (HODs) that had anodes made of pristine or PFSA-doped 4LG (Figure S13a). Because of the large energy difference ($> 1.0 \text{ eV}$) between the highest occupied molecular orbital (HOMO) energy level of the NPB ($\sim 5.4 \text{ eV}$) and WF of pristine graphene ($\sim 4.3 \text{ eV}$), hole injection from the pristine graphene anode is seriously obstructed, so a HOD with the pristine 4LG showed very low current density. When the graphene was doped with PFSA, the hole current density in the HOD was dramatically increased by a factor of $> 10^3$ due to a $\sim 0.8 \text{ eV}$ increase in the surface WF and reduced R_{sh} (Figure S13b). Hole injection efficiencies (η) were calculated, and η of HOD with PFSA-doped graphene was greatly increased by a factor of higher than 10^3 (Figure S13c). We also fabricated green-emitting phosphorescent OLEDs using PFSA-doped graphene to demonstrate its use as an anode. We uniformly spin-coated the polymeric hole-injection layers diluted with IPA on hydrophobic PFSA-doped graphene surface (Figure S14). The OLED with the PFSA-doped graphene anode also exhibited a higher current density than did the OLED that with pristine 4LG anode; this result was also caused by improved hole injection from graphene anode due to improved surface WF of PFSA doped

graphene (Figure S15a). The OLED with PFSA-doped 4LG also showed reduced operating voltage compared to that with pristine 4LG because PFSA-doped graphene anode has lower R_{sh} and higher hole injection capability (Figure S15b). As a result, the device with PFSA-doped 4LG also showed higher current efficiency (CE) and higher power efficiency (PE) (~98.5 cd/A and ~95.6 lm/W without an outcoupling structure) than did the device with the pristine 4LG (~82.7 cd/A and ~77.6 lm/W) (Figure S15c,d). The improved electroluminescent properties of OLED with the PFSA-doped graphene demonstrate the possibility of using PFSA-doped graphene as flexible anode to simultaneously reduce operating voltage and to increase luminous efficiency.”

Reviewer #2

The authors report an extremely environmentally-stable graphene electrode doped with macromolecular acid PFSA as a chemical p-type dopant, resulting in a stretchable, conductive, and transparent materials. The chemically doped graphene to achieve high conductivity have been developed for the decade. However, most works are not stable, thus this work shows high impact in this research field. Overall, the manuscript was well organized and written. Also, the study was well conducted and the discussion is convincing. This referee recommends to accept and publish the article after clarifying some arguments as shown below:

1. There is a similar chemical dopant TFSA, which has similar p-type doping on graphene. Although it is a small molecular, the typical papers should be well cited, such as Nano Lett., 2012, 12 (6), pp 2745–2750, Nanotechnology. 2011 Oct 21;22(42):425701, RSC Adv., 2016, 6, 32746

Answer) We thank the reviewer for suggesting more references. We added [*Nano Lett.* 2012, 12, 2745-2750; *Nanotechnology* 2011, 22, 425701; *RSC. Adv.* 2016, 6, 32746-32756] as references #35, #34, and #29. All the papers the reviewer mentioned were very useful sources of information about various aspects of chemical graphene doping and its applications. We are glad to have these references.

2. In page 4, line 81: ...Since the development of polymeric acid (i.e., poly(styrene81 sulfonic acid)) doping of conducting polymers greatly improved its processability and stability. What're the features when compare the performance and stability of this work with these reported graphene-PSS:PEDOT composites?

Answer) We thank the reviewer for asking this question. Due to the reviewer's question, we can clarify our motivation of this research more clearly. The intention of our statement "Since the development of polymeric acid (*i.e.*, poly(styrene sulfonic acid)) doping of conducting polymers greatly improved its processability and stability," was to explain the development history of conducting polymers. At the beginning stage of research on conducting polymers, inorganic small molecular acids (*e.g.*, HCl, H₂SO₄) and organic small molecular acids (*e.g.*, dodecylbenzenesulfonic acid, camphorsulfonic acid) [*Synth. Met.* 1992, 48, 91] were used as chemical dopants to increase the conductivity of conducting polymers; a similar strategy is used to improve the conductivity of graphene in current graphene research. Even though chemical doping using small-molecular acid improves conductivity of conducting polymers, they still have poor long-term stability and poor film-forming property, so the process has not led to their commercialization. Since the polymeric acid (poly(styrene sulfonic acid), PSS) improved the stability, and film formability, conducting polymers were commercialized, and

are widely used in device applications [*Adv. Mater.* 2000, 12, 481]. The doping history of conducting polymers inspired us to focus on graphene doping using macromolecular dopants.

To study the doping effect of PEDOT:PSS on graphene, we applied PEDOT:PSS on graphene and investigated the electrical and optical properties. PEDOT:PSS were spin-cast on pristine graphene layer without any treatment to make graphene surface hydrophilic. Film uniformity of spin-cast PEDOT:PSS on hydrophobic graphene surface was poor, and left a large portion of uncoated regions (Figure R4a,b). Raman spectra indicated the doping characteristics of graphene/ PEDOT:PSS. There was no clear shift of Raman characteristic bands of the uncoated region (G band of pristine graphene: 1589 cm^{-1} , G band of uncoated region: 1589 cm^{-1}). On the contrary, the coated region showed upshift of G band to 1607 cm^{-1} (Figure R4c); this change indicates that PEDOT:PSS imparted p-type doping on graphene. To investigate the ambient stability of PEDOT:PSS-doped graphene, the Raman spectrum was measured again. PEDOT:PSS-doped graphene was stored in ambient condition for 4 d (Figure R4d). Upshifted G band position was downshifted from 1607 cm^{-1} to 1597 cm^{-1} ; this change indicates that the p-type doping effect had weakened. The change occurs because PSS is hygroscopic, so it takes up moisture from ambient air and thereby degrades the p-type doping effect of PEDOT:PSS on graphene.

Figure R4. Doping characteristics of spin-cast PEDOT:PSS on graphene. Optical microscopy image of (a) pristine graphene, and (b) PEDOT:PSS coated graphene. Scale bar, 100 μm . Raman spectra of (c) different regions marked in (b), and (d) in coated region as prepared and after exposure to ambient conditions for 4 d.

Revised parts in the manuscript)

Page 13 in the manuscript,

“As a control measurement, we compared doping effect of PFSA with those of an inert polymer PMMA which is generally used as a polymer supporter during wet-transfer of graphene, and a conventional conducting polymer PEDOT:PSS which is widely used as a hole-injection material in organic-optoelectronics (Figure S10). In contrast to PFSA, PMMA has neither fluorinated nor acidic groups, so it cannot induce significant p-type doping effect. PMMA provided negligible WF increase and much smaller R_{sh} decrease (~29.4%) compared to that with PFSA (Figure S10). The hydrophilic PEDOT:PSS does not form a uniform film on a graphene surface due to the hydrophobicity of graphene (Figure S11a,b). Because PEDOT:PSS is acidic, the coated region showed G-band upshift (1589 cm^{-1} to 1607 cm^{-1}), which indicates that PEDOT:PSS had a p-type doping effect on graphene (Figure S11c). However, upshifted G-band was downshifted to 1597 cm^{-1} in ambient condition after 4 d; this change indicates that this doping has poor ambient stability. PEDOT:PSS is highly hygroscopic⁶⁰, so it easily takes up moisture in ambient conditions, and thus it has poor doping stability in ambient conditions (Figure S11d).”

3. Page 6 Line 121, weaken the statement “without defect” to “lower defect”. There is no evidence to show this graphene is single crystalline.

Answer) We thank the reviewer for suggesting the clarification of the statement. We revised the expression to comply with the reviewer’s comment as follows.

Revised parts in the manuscript)

Page 6 in the manuscript,

“The spectrum of the pristine graphene showed a very small D-band ($\sim 1350 \text{ cm}^{-1}$), which indicates that high-quality graphene had been successfully grown and transferred onto the substrate with lower defect.”

4. The statement: “... this result proves that solution-processed doping of the PFSA does not induce any significant structural defects in the graphene lattice...”. The authors should provide C1s, S and F1s spectra from XPS to he detail chemical bonding states of C-F or C-S.

Answer) We thank the reviewer for this helpful comment. We performed further X-ray photoelectron spectroscopy (XPS) to analyze the detailed chemical bonding states of PFSA-doped graphene. In contrast to pristine graphene in XPS survey spectrum, S2p, and F1s characteristic peak arises at binding energy of $\sim 170 \text{ eV}$, and $\sim 690 \text{ eV}$, respectively in PFSA-doped graphene (Figure R5a) [*J. Membr. Sci.* 2011, 366, 325]. The C1s spectrum of PFSA-doped graphene revealed C-C sp^2 bonding ($\sim 284.7 \text{ eV}$), with four PFSA-related chemical bondings (*i.e.*, C-O-C ($\sim 286.5 \text{ eV}$), C-S ($\sim 289.6 \text{ eV}$), $-\text{CF}_2$ ($\sim 292.4 \text{ eV}$), $-\text{CF}_3$ ($\sim 294.0 \text{ eV}$), which are components of PFSA [*J. Membr. Sci.* 2011, 366, 325] (Figure R5c,d).

Figure R5. XPS analysis. (a) Survey spectra of pristine and PFSA-doped graphene. (Inset: enlarged S2p peak in survey spectrum of PFSA-doped graphene). (b) Chemical structure of PFSA.

(c) Deconvoluted XPS C1s spectra of PFSA-doped graphene.

Revised parts in the manuscript)

Page 14 in the manuscript,

“By analyzing X-ray photoelectron spectroscopy of p-doped graphene, we also confirmed that PFSA molecule remains on the graphene surface: intense F1s peak (~690 eV), and S2p peak (~170 eV) (Figure S12a). The C1s spectrum of PFSA-doped graphene revealed C-C sp² bonding (~284.7 eV), with four PFSA-related chemical bonds (*i.e.*, C-O-C (~286.5 eV), C-S (~289.6 eV), -CF₂ (~292.4 eV), -CF₃ (~294.0 eV),^{23,62} which can be confirmed in the chemical structure of PFSA (Figure S12b,c).”

Page 15 in the supporting information,

“We performed X-ray photoelectron spectroscopy (XPS) to investigate the change of chemical composition by PFSA doping. Pristine and PFSA-doped graphene showed an O1s peak that could be caused by oxygen in a glass substrate, or PMMA left after the wet-transfer process. PFSA-doped graphene showed an intense F1s peak (~690 eV), and S2p peak (~170 eV) (Figure S12a). The C1s spectrum of PFSA-doped graphene revealed C-C sp² bonding (~284.7 eV), with four PFSA-related chemical bonds (*i.e.*, C-O-C (~286.5 eV), C-S (~289.6 eV), -CF₂ (~292.4 eV), -CF₃ (~294.0 eV),^{23,61} which can be confirmed in the chemical structure of PFSA (Figure S12b,c); these results confirm that a PFSA layer forms on the graphene surface.”

5. The authors used normalized Rsh to define the doping performance over the whole manuscript. However, the value of Rsh of their pristine 1L and 4L graphene should be presented before this normalization. Also, the author find a good approach (c-AFM) to evaluate the uniformity of the doped graphene. However, the AFM is limited due to their quite localized characterization, when considering the large area uniformity, this may not provide additional information. The authors could provide the spatial mapping

of R_{sh} to demonstrate this factor they could provide OLED lighting image to direct evidence this electrical uniformity

Answer) We thank the reviewer for the valuable comments. Here, we present the absolute R_{sh} value as well as its normalized results. To investigate chemical, and thermal stability, we prepared vacuum annealed single-layered graphene on Si/SiO₂ (300 nm) substrate in order to avoid any inevitable side doping effect by PMMA, an etchant, oxygen and water that can occur during wet graphene transfer process. In chemical stability measurement, we used pristine single layer graphenes transferred on Si/SiO₂ substrate, which had $711.5 \pm 155.6 \text{ } \Omega/\text{sq}$ for HNO₃ doping, and $807.5 \pm 149.2 \text{ } \Omega/\text{sq}$ for PFSA doping (Figure R6a). R_{sh} of HNO₃-doped graphene was $221.2 \pm 5.9 \text{ } \Omega/\text{sq}$, and it increased as T_a increased to $423.3 \pm 94.6 \text{ } \Omega/\text{sq}$ by deionized (DI) water, $449.4 \pm 23.0 \text{ } \Omega/\text{sq}$ by isopropyl alcohol (IPA), $346.7 \pm 74.9 \text{ } \Omega/\text{sq}$ by toluene, and $396.1 \pm 15.6 \text{ } \Omega/\text{sq}$ by dimethylsulfoxide (DMSO). However, R_{sh} of PFSA-doped graphene was $372.9 \pm 36.1 \text{ } \Omega/\text{sq}$, and it showed negligible change ($379.8 \pm 13.7 \text{ } \Omega/\text{sq}$ by DI water, $387.0 \pm 14.9 \text{ } \Omega/\text{sq}$ by IPA, $388.8 \pm 14.1 \text{ } \Omega/\text{sq}$ by toluene, and $388.8 \pm 11.2 \text{ } \Omega/\text{sq}$ by DMSO, respectively). In thermal stability, we used pristine graphenes, which had $711.5 \pm 155.6 \text{ } \Omega/\text{sq}$ for HNO₃ doping and $913.2 \pm 115.3 \text{ } \Omega/\text{sq}$ for PFSA doping (Figure R6b). R_{sh} of HNO₃-doped graphene ($215.3 \pm 8.5 \text{ } \Omega/\text{sq}$) increased as T_a increased to $261.3 \pm 26.1 \text{ } \Omega/\text{sq}$ for 100 °C, $329.9 \pm 32.4 \text{ } \Omega/\text{sq}$ for 200 °C, $381.4 \pm 32.7 \text{ } \Omega/\text{sq}$ for 250 °C, and $472.8 \pm 37.2 \text{ } \Omega/\text{sq}$ for 300 °C. In contrast, R_{sh} of PFSA-doped graphene showed R_{sh} decreased to $395.8 \pm 37.8 \text{ } \Omega/\text{sq}$ for 100 °C, $365.6 \pm 37.8 \text{ } \Omega/\text{sq}$ for 200 °C, $352.0 \pm 5.5 \text{ } \Omega/\text{sq}$ for 250 °C, and $337.6 \pm 25.9 \text{ } \Omega/\text{sq}$ for 300 °C. These results indicate that a PFSA-doped graphene had much better chemical and thermal stability than did a HNO₃-doped graphene. To verify the long-term air-stability of PFSA, we monitored the R_{sh} change of PFSA-doped and HNO₃-doped graphenes on glass substrate under ambient condition (Figure R6c). Although HNO₃ doping reduced the

R_{sh} of graphene to $\sim 38.3\%$ of the pristine graphene (Pristine: $539.3 \pm 64.2 \text{ } \Omega/\text{sq}$, and HNO_3 -doped: $206.7 \pm 52.5 \text{ } \Omega/\text{sq}$), R_{sh} increased over 4 d to $\sim 74.1\%$ ($399.7 \pm 104.4 \text{ } \Omega/\text{sq}$) of the pristine graphene. On the contrary, the PFSA-doped graphene on glass substrate showed decreased R_{sh} to $\sim 55.4\%$ compared to that of pristine graphene (Pristine: $518.5 \pm 33.2 \text{ } \Omega/\text{sq}$, and PFSA-doped: $287.3 \pm 18.9 \text{ } \Omega/\text{sq}$); this reduction in R_{sh} had retained for two months (64 d) without noticeable change under ambient conditions.

Figure R6. Stabilities of PFSA and HNO_3 doped graphene. Absolute sheet resistance changes under the (a) various solvent treatments, (b) annealing temperatures, and (c) ambient condition as a function of exposure time. (inset: image of water droplet and contact angle on PFSA and HNO_3 doped graphene)

We prepared four-layered large-area graphene ($> 4 \text{ cm} \times 4 \text{ cm}$) on glass substrate by repeating the wet-transfer (Figure R7a). To prove the uniformity of our PFSA-doping over large-area, we performed spatial mapping of R_{sh} using EddyCus® TF map 2525SR system rather than providing OLED lighting image to directly display the large-area uniformity of PFSA doping (Figure R7b). Pristine four-layered graphene showed $352.7 \pm 48.0 \text{ } \Omega/\text{sq}$. PFSA doping substantially reduces the R_{sh} of graphene throughout the large-area, and PFSA-doping followed by $300 \text{ }^\circ\text{C}$ annealing showed significant R_{sh} reduction with less R_{sh} deviation (Pristine: $352.7 \pm 48.0 \text{ } \Omega/\text{sq}$, PFSA-doped: $91.4 \pm 30.1 \text{ } \Omega/\text{sq}$). It also proves spatial uniformity and excellent doping stability of PFSA-doped graphene.

Figure R7. (a) Large-area ($> 4 \text{ cm} \times 4 \text{ cm}$) four-layered graphene (4LG) transferred on glass substrate, and (b) Spatial R_{sh} mapping of pristine 4LG (left), and PFSA-doped and $300 \text{ }^\circ\text{C}$ annealed 4LG.

Revised parts in the manuscript)

Page 7 in the manuscript,

“HNO₃ caused a stronger doping effect than PFSA: compared to pristine graphene, the R_{sh} of HNO₃-doped single graphene on Si/SiO₂ substrate was reduced by $\sim 68.9 \pm 0.8\%$ (Pristine: $711.5 \pm 155.6 \text{ } \Omega/\text{sq}$, and HNO₃-doped: $221.2 \pm 5.9 \text{ } \Omega/\text{sq}$), whereas the R_{sh} of PFSA-doped graphene on Si/SiO₂ substrate was reduced by $\sim 53.5 \pm 4.9\%$ (Pristine: $807.5 \pm 149.2 \text{ } \Omega/\text{sq}$, and PFSA-doped: $372.9 \pm 36.1 \text{ } \Omega/\text{sq}$). However, when various solvents were spin-cast on the p-doped graphenes (“asD” in Figure 2a), R_{sh} of HNO₃-doped graphene on Si/SiO₂ substrate significantly increased by $\sim 91.4\%$ ($423.3 \pm 94.6 \text{ } \Omega/\text{sq}$) for DI water, $\sim 103.2\%$ ($449.4 \pm 23.0 \text{ } \Omega/\text{sq}$) for IPA, $\sim 56.8\%$ ($346.7 \pm 74.9 \text{ } \Omega/\text{sq}$) for toluene, and $\sim 79.1\%$ ($396.1 \pm 15.6 \text{ } \Omega/\text{sq}$) for DMSO compared to R_{sh} of the as-doped graphene on Si/SiO₂ substrate ($221.2 \pm 5.9 \text{ } \Omega/\text{sq}$, Figure 2a). In contrast, changes of R_{sh} in the PFSA-doped graphene were negligible when it was treated with the solvents (Figure 2a).”

Page 8 in the manuscript,

“The thermal stability of PFSA-doped graphene was also much superior to that of conventional HNO₃-doped graphene. To verify doping stability under high temperature, each p-type doped graphene was thermally annealed at $100 \text{ } ^\circ\text{C} \leq \text{annealing temperature } (T_a) \leq 300 \text{ } ^\circ\text{C}$ in ambient conditions. Compared to R_{sh} of the as-doped graphene, R_{sh} of PFSA-doped graphene on Si/SiO₂ substrate ($435.1 \pm 8.48 \text{ } \Omega/\text{sq}$) gradually decreased as T_a increased, by $\sim 9.03\%$ (to $395.8 \pm 37.8 \text{ } \Omega/\text{sq}$) at $T_a = 100 \text{ } ^\circ\text{C}$, $\sim 16.0\%$ (to $365.6 \pm 37.8 \text{ } \Omega/\text{sq}$) at $T_a = 200 \text{ } ^\circ\text{C}$, $\sim 19.1\%$ (to $352.0 \pm 5.49 \text{ } \Omega/\text{sq}$) at $T_a = 250 \text{ } ^\circ\text{C}$, and $\sim 22.4\%$ (to $337.6 \pm 25.9 \text{ } \Omega/\text{sq}$) at $T_a = 300 \text{ } ^\circ\text{C}$, whereas R_{sh} of HNO₃-doped graphene on Si/SiO₂ substrate ($215.3 \pm 8.49 \text{ } \Omega/\text{sq}$) increased as T_a increased by $\sim 21.4\%$ (to $261.3 \pm 26.1 \text{ } \Omega/\text{sq}$) at $T_a = 100 \text{ } ^\circ\text{C}$, $\sim 53.3\%$ (to 329.9 ± 32.4

Ω/sq) at $T_a = 200$ °C, $\sim 77.2\%$ (to 381.4 ± 32.7 Ω/sq) at $T_a = 250$ °C, and $\sim 119.6\%$ (to 472.8 ± 37.2 Ω/sq) at $T_a = 300$ °C (Figure 2c).”

Page 6 in the manuscript,

“To prove the uniformity of our PFSA-doping over a large area, we prepared 4LG of large-area (> 3 cm \times 4 cm) on glass substrate performed spatial mapping of R_{sh} using EddyCus® TF map 2525SR system (Figure 1e,f). Pristine four-layered graphene without vacuum annealing showed $R_{sh} = 352.7 \pm 48.0$ Ω/sq . PFSA doping substantially reduces the R_{sh} of graphene throughout the large-area, and PFSA-doping followed by 300 °C annealing showed significant R_{sh} reduction with less R_{sh} variation (Pristine: 352.7 ± 48.0 Ω/sq , PFSA-doped: 91.4 ± 30.1 Ω/sq) (Figure 1e,f). This result demonstrates the spatial large-area uniformity of PFSA-doped graphene.”

6. Figure 2 a and b require error bar just like Fig2c.

Answer) After repeating the stability measurements, we were able to achieve statistical results and added error bars to figure 2a, and 2b as the reviewer indicated.

Revised parts in the manuscript)

We revised the figure 2 by adding the relevant error bars.

Revised figure 2,

7. Fig 2 c, the PFSA doped graphene shows the hydrophobic property, which was due to the fluorinated groups. The might be another factor to repulse the foreign molecular and then enhance the chemical stability.

Answer) We agree with the reviewer's comment. Along with non-volatility by macromolecular properties (*i.e.*, non-volatility) of PFSA, its hydrophobicity is another major contributor to its outstanding doping stability on graphene. The molecular structure of PFSA contains affluent fluorinated alkyls (Figure R8a), which significantly reduce the surface energy, so it tends to repulse water and organic molecules. This characteristic can be confirmed by measuring the contact angle on PFSA-doped graphene surface. PFSA-doped graphene had higher contact angle than pristine and HNO₃-doped graphene (Figure R8b-d). Surface energies were calculated from the contact angles of deionized water and

diiodomethane. PFSA has the lowest surface energy among them; PFSA tends to strongly repulse foreign molecules, so it has outstanding chemical stability.

Figure R8. (a) Chemical structure of PFSA. Water and diiodomethane contact angle on (b) pristine, (c) HNO₃-, and (d) PFSA-doped four-layered graphene.

Revised parts in the manuscript)

Page 9 in the manuscript,

“The hydrophobic surface with low surface energy of the PFSA-doped graphene can more stably maintain the p-type doping effect in ambient conditions than can HNO₃-doped 4LG (surface energy: 41.20 mJ/m²). **The outstanding doping stability of PFSA on graphene can be attributed partly to the affluent fluorinated alkyls in PFSA, which substantially reduces the**

surface energy of the PFSA-doped graphene surface; it would repulse the molecules of applied solvents or in ambient air, and maintain the doping effect⁵³.”

8. The authors claim the chemical stability of doped graphene, while the experiential test only organic solvents. Can this dopant sustain well to acid or base chemicals?

Answer) We thank the reviewer for suggesting the new experiments to support our claims. Due to the reviewer’s valuable comment, we found a more strong point to explain the advantage of our macromolecular doping by PFSA. We performed stability test against acid to base chemicals. We dipped HNO₃- and PFSA-doped graphene into solution of strong acid (HCl, $K_a > 1$), weak acid (CH₃COOH, $K_a: \sim 1.8 \times 10^{-5}$), weak base (NH₄OH, $K_b: \sim 1.8 \times 10^{-5}$), or strong base (NaOH, $K_b > 1$) for 15 s, then removed the acid or base residue by blowing N₂. We used pristine graphenes on Si/SiO₂ substrate, which had $700.2 \pm 118.8 \text{ } \Omega/\text{sq}$ for HNO₃ doping, and $807.5 \pm 149.2 \text{ } \Omega/\text{sq}$ for PFSA doping (Figure R8a). R_{sh} of HNO₃-doped graphene on Si/SiO₂ substrate was $229.3 \pm 15.6 \text{ } \Omega/\text{sq}$, and it increased after acid and base treatment (~66.4% ($381.6 \pm 83.0 \text{ } \Omega/\text{sq}$) by HCl, ~118.2% ($500.3 \pm 67.2 \text{ } \Omega/\text{sq}$) by CH₃COOH, ~479.5% ($1329 \pm 131.3 \text{ } \Omega/\text{sq}$) by NH₄OH). Especially, R_{sh} of HNO₃-doped graphene after NaOH treatment was immeasurable by using 4-point probe method (Figure R8a). It can be attributed to tearing of the surface of HNO₃-doped graphene by base chemicals (*i.e.*, NH₄OH, and NaOH) (Figure R8b). In contrast to the HNO₃-doping, uniform surface of PFSA-doped graphene was stably maintained even in strong base solution (*i.e.*, NaOH), resulted in much less R_{sh} increase than that of HNO₃-doped graphene (Figure R8c). This results also proves the chemical stability of PFSA-doping on graphene.

Figure R9. Stabilities of PFSA and HNO₃ doped graphene against acid and base chemicals. (a) Sheet resistance changes under the various acid, and base chemicals (left: normalized, right: absolute value), Optical microscopy image (a) HNO₃-doped graphene, and (b) PFSA-doped graphene (left: before base treatment, right: after base treatment). Scale bar: 500 μm.

Revised parts in the manuscript)

Page 7 in the manuscript,

“We tested chemical stability by dipping HNO₃- and PFSA-doped graphene into a solution of strong acid (HCl, $K_a > 1$), weak acid (CH₃COOH, $K_a: \sim 1.8 \times 10^{-5}$), weak base (NH₄OH, $K_b: \sim 1.8 \times 10^{-5}$), or strong base (NaOH, $K_b > 1$) for 15 s. R_{sh} of HNO₃-doped graphene largely increased after acid and base treatments (Figure 2b). Especially, R_{sh} of HNO₃-doped graphene after NaOH treatment was unmeasurable using the 4-point probe method; this result can be attributed that the surface of HNO₃-doped graphene was torn after treatment with base chemicals (Figure S1a). In contrast, the surface of PFSA-doped graphene remained smooth even in strong base solution, so its R_{sh} increased much less than it did in HNO₃-doped graphene (Figure S1b). This result demonstrates that of p-type doping using macromolecular acid is stable against almost every chemical environments (e.g., non-polar, polar aprotic, polar protic solvents, acid and base solution).”

Page 2 in the supporting information,

“We performed stability test against chemicals from acid to base. We dipped HNO₃- and PFSA-doped graphene into the solution of strong acid (HCl, $K_a > 1$), weak acid (CH₃COOH, $K_a: \sim 1.8 \times 10^{-5}$), weak base (NH₄OH, $K_b: \sim 1.8 \times 10^{-5}$), or strong base (NaOH, $K_b > 1$) for 15 s, then removed the acid or base residue by blowing N₂. The surfaces of the HNO₃-doped graphene were torn by surface treatment with base chemicals (Figure S1a); in contrast, the surfaces of PFSA-doped graphene were much less affected even by strong base solution, and resulted in much less R_{sh} increase in HNO₃-doped graphene (Figure S1b). This result also proves the chemical stability of PFSA-doping on graphene.”

9. Line 198, correct to “10um²x10um².”

Answer) We revised our manuscript as the reviewer indicated.

Revised parts in the manuscript)

Page 10 in the manuscript,

“To monitor changes in n and strain as a function of T_a after PFSA doping, we obtained a hundred Raman spectra for each graphene by raster-scanning area of $10\ \mu\text{m} \times 10\ \mu\text{m}$,”

Page 6 in the manuscript,

“All surface topographic images were measured over $5\ \mu\text{m} \times 5\ \mu\text{m}$ of the graphene surface.”

10. Line 201, the concentration (n) of doped graphene is~ 2 times higher than that of pristine graphene. Normally it should be 1 or 2 order increase to achieve significant WF shifting and Rsh reduction. Can it be well explained? Will it be better to employ the hall measurement to probe this N? The Raman single may also effect by the underlying substrate (localized strain).

Answer) We thank the reviewer for the constructive question. Our PFSA doping exhibited ~55% reduction of R_{sh} , ~0.8 eV increase of WF , and hole concentration of graphene was calculated from Raman spectroscopy, and it was increased from $\sim 3.1 \times 10^{12}$ (Pristine) to ~ 1.1

$\times 10^{13}$ (PFSA-doped) (Figure R10a). Fermi energy change (δE_F) is proportional to \sqrt{n} , where n denotes hole concentration (Figure R10b) [Phys. Rev. Lett. 2008, 101, 136804]. Using the relation, δE_F by PFSA doping is calculated as ~ 200 meV (black dashed line). Using Dirac point shift from current-voltage characteristics of graphene-FETs, hole concentration of graphene can also be calculated. Hole concentration of pristine graphene was $\sim 4.46 \times 10^{12} \text{ cm}^{-2}$, and it was increased to $> 1.8 \times 10^{13} \text{ cm}^{-2}$, which corresponds to > 500 meV increase of δE_F . Considering the hole concentration and δE_F from FET results, the increase of hole concentration is substantial, as the reviewer indicated.

Figure R10. Calculated (a) hole concentration from Raman spectroscopy (black), and current-voltage characteristics of FETs (red), and (b) Fermi energy shift with respect to the calculated charge concentration (black, and red dashed line corresponds to the results from Raman spectroscopy, and current-voltage characteristics of FETs, respectively) [Phys. Rev. Lett. 2008, 101, 136804]

In contrast to HNO_3 , which has an in-plane molecular dipole moment, the simplest PFSA has non-planar dipole moment (Figure R11). The real molecular structure of the extended fluorinated alkyl chain that corresponds to the real chemical structure of PFSA, would have

more substantial non-planar dipole moment than that of the simplest PFSA, and this dipole can attribute to the interfacial dipole on graphene. Therefore, a substantial interfacial dipole is formed on graphene surface; the dipole induces a significant shift of vacuum level. Due to these synergistic effects of δE_F by increase of hole concentration, and vacuum level shift from interfacial dipole formation, PFSA doping resulted in substantial WF (~ 0.8 eV) increase.

Figure R11. Molecular configuration of (a) HNO_3 , and (b) PFSA. Spheres: blue = oxygen, silver = sulfur, cyan = hydrogen, orange = fluorine, yellow = carbon. Blue arrow: direction of molecular dipole.

Thermal treatment of graphene generally induces strain, because of differences in the thermal expansion coefficients of graphene and its substrate. Epitaxially-grown CVD-graphene using SiC substrate at high temperature (> 1100 °C) showed compressive strain ($\sim 1\%$) at room temperature [Nano Lett. 2008, 8, 4320-4325]. Annealing $\text{SiO}_2/\text{graphene}$ at 300 °C causes formation of sub-nanometer-height ripples in graphene, which are evidence of induced strain [Nat. Commun. 2012, 3, 1024]. In our work, we distinguished the hole-doping effect from the strain effect. Considering the variations in the characteristic bands 2D and G Raman bands of pristine and PFSA-doped graphene (Figure 3a), annealed PFSA-doped

graphene showed slight increase of strain compared with pristine graphene (Pristine graphene: 0.1% to -0.2%, PFSA-doped graphene annealed at 300 °C: 0.1% to -0.3%); the difference can be attributed to thermal deformation during annealing.

Revised parts in the manuscript)

Page 10 in the manuscript,

“Notably, PFSA-doped graphene treated at $T_a = 300$ °C showed $n \sim 11 \times 10^{12} \text{ cm}^{-2}$, which is three times higher than n in a pristine graphene (Figure 3b,c). **PFSA-doped graphene showed slight increase of strain (Pristine graphene: 0.1% to -0.2%, PFSA-doped graphene at $T_a = 300$ °C: 0.1% to -0.3%); this change can be attributed to slight thermal deformation during thermal annealing.**”

11. Line 210, the statement” this also indicates that sulfonic acid groups (-SO₃H) would be rearranged towards underlying graphene, and that thereby cause increase in n of PFSA-doped graphene.” The authors provide a good DFT calculation to evidence this argument. However, this argument can be directly evidenced by c-AFM to understanding this potential difference before or after 300oC treatment.

Answer) We thank the reviewer for the constructive comment. After thermal annealing, the water contact angle on PFSA film substantially increased, which strongly indicates that substantial rearrangement occurs in the surface of PFSA film [ECS Trans. 2012, 50, 951-959]. DFT calculation suggested that p-type doping of PFSA was dominated by the electronic

dipole interaction between graphene and $-\text{SO}_3\text{H}$ in PFSA. To experimentally support the DFT results, we investigated the p-type doping effect by measuring spatial R_{sh} , with surface potential to evidence the $-\text{SO}_3\text{H}$ rearrangement as T_a increased. R_{sh} of pristine graphene decreased as T_a of PFSA increased (Figure R12a-d). Also, current-voltage characteristics of graphene-based FETs indicates that increase in T_a induces upshift of the Dirac point (Figure R12e). Furthermore, WF increased as T_a increases. This relationship also suggests an increase in the p-type doping effect or interfacial dipole on graphene, and is further evidence of surface rearrangement. These results prove the intensive p-type doping effect of thermal annealing. Overall, thermal annealing induced rearrangement in PFSA film, and $-\text{SO}_3\text{H}$ in PFSA moved towards the graphene; this migration strengthened the p-type doping effect of PFSA on graphene.

Figure R12. Large-area spatial (30 mm \times 45 mm) R_{sh} mapping of (a) pristine, and PFSA-doped followed by thermal annealing at (b) 100 °C, (c) 200 °C, (d) 300 °C. (e) Current-voltage characteristics of FETs with pristine graphene (black), and PFSA-doped and annealed at 100 °C (red), 200 °C (green), or 300 °C (blue). (f) Graphene WF of pristine (black), and PFSA-doped and annealed at 100 °C (red), 200 °C (green), or 300 °C (blue). (inset: potential mapping results from Kelvin probe SKP-5050 measurement system)

Revised parts in the manuscript)

Page 12 in the manuscript,

“This result also implies that rearrangement of PFSA at high T_a influences the n of PFSA-doped graphene because the number of acidic protons in the proximity of graphene increases.

Spatial R_{sh} mapping (Figure S7a-d), Raman spectroscopy (Figure 3a), WF (Figure 4d), and FET results (Figure S5a) of pristine, and PFSA-doped with subsequent thermal annealing with increased T_a showed sequential R_{sh} decrease and increased n as T_a increased; these results are direct evidence that increased T_a influences the doping effect of PFSA (*i.e.*, increase of acidic protons in the proximity of graphene).”

Page 8 in the supporting information,

“To experimentally support the DFT results, we investigate the p-type doping effect by measuring spatial R_{sh} mapping. R_{sh} of pristine graphene sequentially decreased as T_a of PFSA increased (Figure S7a-d).”

12. Line 287. Since the doped graphene shows highly hydrophobic surface. How the stacking film can be uniform formed by coating hydrophilic PEDOT:PSS on doped graphene?

Answer) As in our previous research [*Nanotechnology* 2014, 25, 014012], when we deposited polymeric hole-injection layers (HILs), we used perfluorinated ionomer (PFI)-blended PEDOT:PSS solution, which contains isopropyl alcohol (IPA) rather than as-purchased PEDOT:PSS. Similar to previous research [*Nanotechnology* 2014, 25, 014012; *Sci. Rep.* 2013, 3, 1581], PEDOT:PSS is not uniformly coated on a hydrophobic graphene surface (Figure R13a). Diluting with IPA significantly reduced contact angle of polymeric solution on graphene surface, because the lower polarity of IPA than that of H_2O improved the wettability of the solution. However, diluting with IPA was still insufficient to form uniform

polymeric HILs on graphene surface. By PFI blending as well as adding IPA to PEDOT:PSS solution we could cast PEDOT:PSS film uniformly on a hydrophobic graphene surface. We also performed mild UV-O₃ treatment to make graphene surface hydrophilic before spin-casting the PFI-blended PEDOT:PSS solution. These attempts resulted in a uniform polymeric HIL film on the graphene electrode (Figure R13b). PFSA-doped graphene has even lower surface energy (Pristine: 55.73 mJ/m², PFSA-doped: 21.69 mJ/m²), resulted in poor film-formability (Figure R13c). However, improved wettability from IPA dilution, and PFI blending on the PFSA-doped graphene, resulted in formation of uniform polymeric HILs on the PFSA-doped graphene surface (Figure R13d).

Figure R13. Optical microscopy images of (a) PEDOT:PSS on graphene, (b) GraHIL on graphene, (c) PEDOT:PSS on PFSA-doped graphene, and (d) GraHIL on PFSA-doped graphene (scale bar, 500 μm).

Revised parts in the manuscript)

Page 15 in the manuscript,

“We uniformly spin-coated the polymeric hole-injection layers diluted with IPA on hydrophobic PFSA-doped graphene surface (Figure S14).”

Page 17 in the supporting information,

“To deposit polymeric hole-injection layers (HILs)^{S4}, we used perfluorinated ionomer (PFI)-blended PEDOT:PSS solution (we call GraHIL), which contains isopropyl alcohol (IPA), rather than as-purchased PEDOT:PSS solution, because it cannot be uniformly coated on hydrophobic graphene surface (Figure S14a). The OLEDs were fabricated with a GraHIL, which has high surface $WF \sim 5.95 \text{ eV}$.^{S5,S6,S7} Diluting with IPA significantly reduced the contact angle of polymeric solution on graphene surface, because the lower polarity of IPA than that of H_2O improved the wettability. However, diluting with IPA was still insufficient to form uniform polymeric HILs on graphene surface. Blending with PFI as well as addition of IPA to PEDOT:PSS solution yielded a uniformly cast PEDOT:PSS film on hydrophobic graphene surface. To make the graphene surface hydrophilic, we also performed mild UV-O_3 treatment before spin-casting the PFI-blended PEDOT:PSS solution. These attempts resulted in a uniform polymeric HIL film on graphene electrode (Figure S14b). PFSA-doped graphene has even lower surface energy (Pristine: 55.73 mJ/m^2 , PFSA-doped: 21.69 mJ/m^2), resulted in

poor film-formability (Figure S14c). However, improved wettability from IPA dilution, and PFI blending works on PFSA-doped graphene, and resulted in formation of uniform polymeric HILs on PFSA-doped graphene surface (Figure S14d).”

Reviewer #3

Authors investigated the effect of PFSA doping on graphene. The interesting points are high p-type doping stability under ambient atmosphere and at elevated temperature up to 300 °C. The sheet resistance of p-doped graphene decreased as a function of annealing temperature, which is strikingly different with typical small molecule doping. By changing work function caused by p-type doping, the efficiency of OLEDs are much improved. Therefore, I recommend that this paper will be published in Nature Commun. after minor revision.

Comments:

1. The substrate for this study is not easy to be recognized. What kind of substrate is used for doping study?

Answer) We thank the reviewer for the meticulous comment. In this work, we used various substrate (*i.e.*, Si/SiO₂ (~300 nm), or glass substrate). Before measuring the stability, we performed vacuum annealing on transferred graphene up to ~500 °C, so we used not a flexible substrate (e.g., polyethylene terephthalate (PET)), but rigid substrate, which can endure such high temperature. When quantifying large-area uniformity and ambient stability, we used glass as a substrate for 4-layered graphene. Contact angle measurement was also performed on glass substrate. For other analyses of doping effect (*e.g.*, Raman spectroscopy,

chemical stability, thermal stability), we used Si/SiO₂ (~300 nm) substrate, which is a widely-used substrate of graphene.

Revised parts in the manuscript)

Page 17 in the manuscript,

“Floated single layer graphene was transferred onto the target substrate. Glass substrates are used to investigate the optical transparency, large-area uniformity, and ambient stability. To perform c-AFM, we used native Si as substrates. In the other characterizations of PFSA doping effect (*e.g.*, Raman spectroscopy, chemical-/thermal- stability test), we used Si/SiO₂ (~300 nm) substrates.”

2. Figure quality should be improved. For instance, The labels in x axis do not look good in Figure 3c and d.

Answer) We thank the reviewer for the constructive comment. We revised these labels, and also revised other figures to improve their quality.

Revised figures are shown below.

Figure 1.

Figure 2

Figure 3

Figure 4.

Reviewers' comments:

Reviewer #1 (Remarks to the Author):

The authors have made significant changes to the paper's content and the claims are better substantiated and convincing. I therefore suggest publication of the work after improvements to the legibility of the paper are made:

1. The paper is too long in light of the relatively simple story and several parts should be shortened.
 - a. The introduction should focus on previous work on stability and achievable doping levels rather than list all previously employed dopants and the history of polymer doping.
 - b. The results can be limited to PFSA with control experiments being mentioned in passing and expanded in the supplementary information.
 - c. The "Discussion" part is more of an application and should be mentioned in a paragraph rather than a page.
2. Conclusions should be given at the end of the paper instead of repeated several times. What is the origin of the increase in doping concentration with annealing temperature?
3. The manuscript was tedious to read which does not do justice to the straightforward content. I would therefore suggest grouping by topics rather than techniques. One possible outline could be: Morphology (AFM, XPS, Raman D-Band, resistance mapping, Kelvin probe mapping), enhanced doping stability (Raman, FET, KPM, resistance), mechanism (contact angle, DFT, annealing temperature effect), application (LED).
4. Numbers should be used sparingly in the text and instead be compiled in supplementary tables.
5. The quality of the graphs has to be improved
 - a. Numerical values should not be on categorical axes.
 - b. Figure 2(a) should be simplified.
 - c. Colors should contribute to the meaning.
6. The newly added content deserves more emphasis in the figures and I think the supplementary figures S9a,c , S5a,d, and S15b should find a place in the main text. To make space, Figures 1(a-c), 4(b,c) could be moved to the supplementary material.
7. Several English mistakes should be fixed (graphenes, lower defects)

Reviewer #2 (Remarks to the Author):

The authors have well-replied this referee's comments point-by-point. However, here raising another question: Is there any correlation between the PFSA thickness and carrier concentration(or sheet resistance)? Now the suggested thickness is ~ 3.4 nm. It's crucial to know this thickness effect on the performance/properties of their conductive film(optical transparency, carrier concentration sheet resistance etc). Also, the pristine PFSA also shows electrical conductivity(dried phase) in nature. Did authors also test this sample as a reference?

Reviewer #3 (Remarks to the Author):

The paper entitled "Extremely stable graphene electrodes doped with macromolecular acid" reported PFSA as a p-type dopant for graphene with high thermal stability. Following the reviewer's questions and suggestions, authors fastidiously reply and modify the manuscript. Therefore, I think the manuscript is suitable for publish on Nature Communications without additional revision.

Response to Reviewers' comments

We appreciate the reviewers' valuable comments. We revised our manuscript to comply with all of them, and have prepared point-by-point responses, which we present here. The revised parts in the manuscript are marked in red.

Reviewer #1

Comments:

The authors have made significant changes to the paper's content and the claims are better substantiated and convincing. I therefore suggest publication of the work after improvements to the legibility of the paper are made:

- 1. The paper is too long in light of the relatively simple story and several parts should be shortened.**
 - a. The introduction should focus on previous work on stability and achievable doping levels rather than list all previously employed dopants and the history of polymer doping.**
 - b. The results can be limited to PFSA with control experiments being mentioned in passing and expanded in the supplementary information.**
 - c. The "Discussion" part is more of an application and should be mentioned in a paragraph rather than a page.**

Response) We appreciate the reviewer’s constructive comments. We revised our manuscript as the reviewer indicated:

(a) We removed the history of conducting polymer doping from the introduction, and focused on stability and achievable doping levels, (b) we present the control measurements of graphene doped with PMMA, and PEDOT:PSS in the supplementary information, and (c) the application mentioned in the discussion has been shortened to a paragraph.

Revised parts in the manuscript)

Page 3 in the manuscript (Introduction),

“Graphene has outstanding electrical, mechanical and optical properties¹⁻⁵, so it has been regarded as an alternative to indium tin oxide (ITO), which is the conventional transparent electrode in optoelectronic devices, but is not suitable for electrodes in flexible optoelectronics due to its brittleness and increasing cost^{6,7}. Since the development of chemical vapor deposition methods to produce high-quality and large-area graphene²⁻⁵, much research has been devoted to applying graphene in flexible electronics such as organic light-emitting diodes (OLEDs)⁸⁻¹⁶, organic solar cells¹⁷⁻¹⁹, and organic transistors²⁰⁻²². However, pristine graphene has high sheet resistance ($R_{sh} > 300 \Omega/\text{sq}$) and low work function ($WF \sim 4.4 \text{ eV}$) which are still inferior to those of ITO ($R_{sh} \sim 10 \Omega/\text{sq}$, $WF \sim 4.8 \text{ eV}$), so pristine graphene must be modified before it can be a practical replacement for ITO electrodes^{23,24}.

Various researchers had used chemical doping to control the electrical properties of pristine graphene^{3,8,14,25-37}. Dopants used in graphene for electrodes to date have been mainly classified into two types: (1) small molecules^{3,8,25-35}, and (2) transition metal oxides^{14,36,37}; both exploit charge transfer on the graphene surface. Charge-transfer doping of graphene

with small-molecule dopants such as inorganic small-molecule acids (e.g., HNO₃, HCl, H₂SO₄)^{3,8,25} and metal chlorides (e.g. AuCl₃, FeCl₃)^{8,26-29} has been widely used and developed to increase the electrical conductivity of graphene; however, graphene that is doped with inorganic small-molecule acid has serious environmental instability, which has been considered as a great impediment to practical application of graphene electrodes. R_{sh} of an HNO₃-doped graphene gradually increases under ambient conditions due to the high volatility of small-molecule acids, and high temperature during device fabrication accelerates significant degradation in its electrical conductivity⁸. After doping with metal chlorides, reduction of metal cations can produce metal particles on the graphene surface; these have two deleterious effects: they can decrease the optical transmittance (OT) of the graphene, and if they are large, they protrude and provide leakage paths for electrical current in thin-film devices^{8,26-29}. Also, transition metal oxide is not uniformly deposited on the graphene surface because the graphene lacks dangling bonds or surface functional groups. Therefore, thermal evaporation of transition metal oxide on graphene also roughens its surface³⁸. Ideal chemical doping of graphene for practical use as an anode in electronic devices should achieve: (1) low R_{sh} , (2) high WF , (3) high stability against heat, chemical, and ambient conditions (4) smooth film surface and (5) high OT .

Here, we introduce a novel approach to macromolecular chemical doping that uses a polymeric acid, which has not been used for graphene doping and demonstrated extraordinary results on the all kinds of aspects of stability (high temperature, chemicals, and air) and high WF ; these characteristics are almost unattainable with conventional small-molecule acid doping. We used a perfluorinated polymeric sulfonic acid (PFSA) for application to flexible graphene anodes in optoelectronics. PFSA is composed of a perfluorinated carbon backbone and sulfonic acid groups. Due to the electron-withdrawing properties from electronic dipole of acidic proton in sulfonic acid groups, PFSA induces p-type doping of graphene.

Furthermore, the perfluorinated carbon backbones have large ionization potential, which substantially increases the surface potential of graphene. Superior thermal and chemical stability of PFSA molecule provided outstanding stability of chemically-doped graphene against high temperature heating and exposure to various chemicals including strong acids and bases^{39,40}. Therefore, the PFSA is an ideal form of macromolecular p-type dopant for graphene electrode. Our unconventional approach of using polymeric acid may stimulate research into high-stability graphene doping, and can be a starting point in development of polymeric-acid dopants for ideal graphene electrodes.”

Page 12 in the manuscript,

“Polymeric graphene doping was performed using other macromolecules (PMMA and PEDOT:PSS), which showed weak p-type doping effect and poor ambient stability, respectively (Figures S14, S15); these results exhibit the uniqueness of the graphene doping using PFSA.”

Page 13 in the manuscript,

“Discussion

We used a macromolecular fluorinated acid, PFSA, as a chemical p-type dopant to give extremely stable chemical p-type doping for graphene. The PFSA-doped graphene met the requirements for ideal p-type doping of graphene anode: (1) large R_{sh} decrease, (2) substantial increase in surface WF , (3) high stability against all kinds of circumstances (high temperatures, chemicals and ambient conditions) (4) smooth and uniform surface without any defects or large particles, and (5) negligible decrease in OT . The non-volatility, strong

binding to graphene, and chemical and thermal stability of PFSA can explain the excellent environmental stability of PFSA-doped graphene.

We also fabricated HODs and OLEDs to demonstrate the superior hole injection and electroluminescent characteristics of devices that used the PFSA-doped graphene as an anode. An HOD that used the PFSA-doped graphene showed dramatic improvement of hole current, and OLEDs that used PFSA-doped graphene exhibited increase of luminous efficiencies: this result demonstrates the possibility of practical anode application of the PFSA-doped graphene due to its improved electrical conductivity and surface WF , and confirms that our PFSA is a promising p-type chemical dopant to make more ideal flexible graphene electrodes with excellent stability against high temperature, chemicals, and ambient conditions, high WF ; simultaneous achievement of these attributes has been almost impossible using conventional doping with small molecules.

This work provides a promising way to overcome the demerits of chemically-doped graphene electrodes and is a significant step toward development of stable graphene electrodes for practical use in various opto-electronic devices.”

2. Conclusions should be given at the end of the paper instead of repeated several times.

What is the origin of the increase in doping concentration with annealing temperature?

Response) Thermal annealing would cause rearrangement of PFSA at high annealing temperature (T_a). Thermal annealing causes the perfluorinated backbone that has low surface energy to become dominant at the film surface, and sulfonic acid groups ($-\text{SO}_3\text{H}$) in PFSA rearrange towards the underlying graphene. By performing density-functional theory calculation on PFSA-doped graphene, we confirmed that the acidic proton in the $-\text{SO}_3\text{H}$ causes electronic dipole interaction with graphene, thereby reducing the electron density of graphene. Rearrangement of PFSA at high T_a influences the hole concentration of PFSA-

doped graphene because the number of acidic protons in the proximity of graphene increases. Therefore, we can conclude that high T_a heating of PFSA-doped graphene strengthens the p-type doping effect in PFSA-doped graphene. To avoid the duplication of conclusion that explains the origin of increase in hole concentration of PFSA-doped graphene, we mentioned it in the part that discusses “Thermal stabilities and doping mechanism of graphene doped with macromolecular fluorinated acid”.

Revised parts in the manuscript)

Page 8 in the manuscript,

“These results also support that rearrangement of PFSA at high T_a influences the n of PFSA-doped graphene because the number of acidic protons in the proximity of graphene increases. All measurements showed that thermal annealing caused R_{sh} decrease, WF increase and n increase in PFSA-doped graphene as T_a increased; these results are direct evidence that increased T_a influences the doping effect of PFSA.”

3. The manuscript was tedious to read which does not do justice to the straightforward content. I would therefore suggest grouping by topics rather than techniques. One possible outline could be: Morphology (AFM, XPS, Raman D-Band, resistance mapping, Kelvin probe mapping), enhanced doping stability (Raman, FET, KPM, resistance), mechanism (contact angle, DFT, annealing temperature effect), application (LED).

Response) To improve the manuscript legible and straightforward for readers, we classified the experimental results into three parts (*i.e.*, (i) Doping effect of a macromolecular

fluorinated acid on graphene (Figure 1), (ii) Thermal stabilities and doping mechanism (Figure 2, and 3), (iii) Chemical and ambient stability (Figure 4), and (iv) Optoelectronics application (Figure 5)) and whole manuscript has been revised following the changed sub-headings.

4. Numbers should be used sparingly in the text and instead be compiled in supplementary tables.

Response) We appreciate a constructive comment of the reviewer. We used tables (**Table S1**, **Table S2**, and **Table S3**) to show the sheet resistance change of p-type doped graphene.

Table S1. R_{sh} change of PFSA and HNO₃-doped graphene against various annealing temperature

		Thermal treatments				
		As doped	100 °C	200 °C	250 °C	300 °C
PFSA	R_{sh}	435.1	395.8	365.6	352.0	337.6
	[Ω/sq]	± 8.48	± 37.8	± 37.8	± 5.49	± 25.9
	R_{sh} increase [%]	-	-9.03	-16.0	-19.1	-22.4
HNO₃	R_{sh}	215.3	261.3	329.9	381.4	472.8
	[Ω/sq]	± 8.49	± 26.1	± 32.4	± 32.7	± 37.2
	R_{sh} increase [%]	-	21.4	53.3	77.2	119.6

R_{sh} increase is calculated as: $(R_{sh} - R_{sh,asD})/R_{sh,asD}$. R_{sh} of pristine single-layered graphene samples, samples, which were doped with HNO₃ and PFSA was 711.5 ± 155.6 Ω/sq, and 913.2 ± 153.3 Ω/sq, respectively.

Table S2. R_{sh} change of PFSA and HNO₃-doped graphene against various solvent treatments

		Chemical solvent treatments				
		As doped	DI water	IPA	toluene	DMSO
PFSA	R_{sh}	372.9	379.8	387.0	388.8	388.8
	[Ω /sq]	± 36.1	± 13.7	± 14.9	± 14.1	± 11.3
	R_{sh} increase [%]	-	4.75	6.75	7.25	7.25
HNO ₃	R_{sh}	221.2	423.3	449.4	346.7	396.1
	[Ω /sq]	± 5.9	± 94.6	± 23.0	± 74.9	± 15.6
	R_{sh} increase [%]	-	91.4	103.2	56.8	79.1

R_{sh} increase is calculated as: $(R_{sh} - R_{sh,asD})/R_{sh,asD}$. R_{sh} of pristine single-layered graphene samples, which were doped with HNO₃ and PFSA was $711.5 \pm 155.6 \Omega$ /sq, and $807.5 \pm 149.2 \Omega$ /sq, respectively.

Table S3. R_{sh} change of PFSA and HNO₃-doped graphene against various acid and bases

		Acid and base treatments				
		As doped	HCl	CH ₃ COOH	NH ₄ OH	NaOH
PFSA	R_{sh}	361.9	370.7	426.0	424.2	629.9
	[Ω /sq]	± 33.2	± 15.8	± 38.3	± 46.7	± 17.8
	R_{sh} increase [%]	-	2.4	17.7	17.2	74.1
HNO ₃	R_{sh}	229.3	381.6	500.3	1329	-
	[Ω /sq]	± 15.6	± 83.0	± 67.2	± 131.3	
	R_{sh} increase [%]	-	66.4	118.2	479.5	-

R_{sh} increase is calculated as: $(R_{sh} - R_{sh,asD})/R_{sh,asD}$. R_{sh} of pristine single-layered graphene samples, which were doped with HNO₃ and PFSA was $700.2 \pm 118.8 \Omega$ /sq, and $807.5 \pm 149.2 \Omega$ /sq, respectively.

Revised parts in the manuscript)

“High-temperature annealing affected R_{sh} (**Table S1**). Compared to R_{sh} of the as-doped graphene, R_{sh} of PFSA-doped graphene on Si/SiO₂ substrate gradually decreased as T_a increased (up to ~22.4% decrease of R_{sh} (asD) at $T_a = 300$ °C), whereas R_{sh} of HNO₃-doped graphene on Si/SiO₂ substrate increased as T_a increased (~119.6% increase of R_{sh} (asD) at $T_a = 300$ °C) (Figure 2a); this trend is due to the high volatility of HNO₃. These results indicate that thermal stability is much better in PFSA-doped graphene than in HNO₃-doped graphene.”

Page 9 in the manuscript,

“To demonstrate the chemical invulnerability of PFSA-doped graphene, polar protic (deionized water, isopropyl alcohol), polar aprotic (dimethyl sulfoxide), and non-polar solvent (toluene) were spin-coated on p-doped **graphene samples**, which were then gently annealed to remove residual solvents. All treatments significantly increased the R_{sh} of HNO₃-doped graphene on Si/SiO₂ substrate, but had negligible effect on R_{sh} in the PFSA-doped graphene (**Table S2**, Figure 4a).”

Page 10 in the manuscript,

“We also tested chemical stability by dipping HNO₃-doped or PFSA-doped graphene into a solution of strong acid (hydrochloric acid, $K_a > 1$), weak acid (acetic acid, $K_a: \sim 1.8 \times 10^{-5}$), weak base (ammonium hydroxide, $K_b: \sim 1.8 \times 10^{-5}$), or strong base (sodium hydroxide, $K_b > 1$) for 15 s. R_{sh} of HNO₃-doped graphene greatly increased after acid and base treatments (Figure 2b, **Table S3**).”

5. The quality of the graphs has to be improved:

a. Numerical values should not be on categorical axes.

b. Figure 2(a) should be simplified.

c. Colors should contribute to the meaning.

Response) We improved the figure quality as the reviewer commented as follows.

a. We removed numerical values of categorical axes in figure 2c (Figure R1).

b. We abbreviated the names of chemical solvents, acids and bases (*i.e.*, DI water → DI, Isopropyl alcohol → IP, Dimethylsulfoxide → DM, and Toluene → TO) as well as (*i.e.*, Hydrochloric acid (HCl) → HA, Acetic acid (CH₃COOH) → AA, Ammonium hydroxide (NH₄OH) → AH, Sodium hydroxide (NaOH) → SH) (Figure R2). We used these abbreviation in categorical axes in Figure 4a,b.

c. We revised the figures so that each color contributed meaning in the manuscript. We set blue as PFSA-doped graphene, and pink as HNO₃-doped graphene. After thermal annealing on PFSA-doped graphene, we set bluish colors on thermally annealed PFSA-doped graphene (*i.e.*, dark cyan for 200 °C, and violet for 300 °C).

Figure R1. Sheet resistance changes against various annealing temperatures

Figure R2. Sheet resistance changes against (a) chemical solvents, and (b) acids and bases.

6. The newly added content deserves more emphasis in the figures and I think the supplementary figures S9a,c , S5a,d, and S15b should find a place in the main text. To make space, Figures 1(a-c), 4(b,c) could be moved to the supplementary material.

Response) We revised the figures as the reviewer indicated. We added supplementary figures (Figures S9a,c, S5a,d, S15b) and have revised the figures in the manuscript as follows:

Figure 1. Macromolecular doping concept and its characteristics on graphene. (a) Chemical structure of perfluorinated polymeric sulfonic acid (PFSA), and schematic drawings of graphene doped using PFSA. (b) Optical transmittance of pristine and PFSA-doped four-layer graphene (inset: large-area transferred four-layered graphene on glass substrate). (c) Raman spectra of pristine and PFSA-doped graphene. Spatial sheet resistance map of (d) pristine 4LG, and (e) PFSA-doped 4LG.

Figure 2. Temperature dependency and doping mechanism. (a) Sheet resistance changes, and (b) potential mapping of pristine graphene (black), PFSA-doped and annealed at 100 °C (blue), 200 °C (dark cyan), or 300 °C (violet) under the various annealing temperatures. (c) Calculated electrostatic potential of the most stable configuration of PFSA molecule (inset: difference in work function between pristine and PFSA-doped graphene), (d) Configuration of PFSA-doped graphene (top) top view, and (bottom) side view. Spheres: blue = oxygen, silver = sulfur, cyan = hydrogen, orange = fluorine, yellow = carbon. The isosurface of the charge density distribution difference is also shown in the figure. Isosurface level is $0.005e/\text{\AA}^3$.

Figure 3. Hole concentration changes of PFSA-doped graphene induced by thermal annealing. (a) Variations in 2D and G Raman bands of HNO₃-doped graphene. (b) Hole concentration histogram, and (c) averaged hole concentration of thermally annealed pristine and PFSA-doped graphene with various annealing temperature calculated from Raman spectroscopy results. (d) Current versus voltage characteristics (inset: optical microscopy image of graphene-based FET (scale bar, 200 μm)), and (e) hole concentrations of pristine graphene (black), PFSA-doped graphene and 100 °C annealed (blue), 200 °C annealed (dark cyan), and 300 °C annealed (violet) calculated from Raman spectroscopy results.

Figure 4. Chemical and ambient stability. R_{sh} changes under the (a) various solvent treatments, (b) acids and bases treatments, and (c) ambient condition as a function of exposure time (inset: image of water droplet and contact angle on PFSA and HNO₃ doped graphene). WF changes under the ambient exposure of (d) PFSA-doped graphene, and (e) HNO₃-doped graphene.

Figure 5. Optoelectronics applications. (a) Schematic of HODs, and OLEDs, (b) Current densities versus voltages of HODs with pristine and PFSA-doped graphene anodes. (c) Luminances versus voltage, and (d) current efficiencies versus luminances of green phosphorescent OLEDs with pristine and PFSA-doped graphene anodes.

7. Several English mistakes should be fixed (graphenes, lower defects).

Response) We corrected the English mistakes; *e.g.*, “graphenes” to “graphene samples”, and “lower defects” to “few defects”.

“To demonstrate the chemical invulnerability of PFSA-doped graphene, polar protic (deionized water, isopropyl alcohol), polar aprotic (dimethyl sulfoxide), and non-polar solvent (toluene) were spin-coated on p-doped **graphene samples**, which were then gently annealed to remove residual solvents.”

Page 10 in the manuscript,

“Before the investigation of graphene’s doping stability, pristine **graphene samples** were vacuum-annealed at 500 °C to eliminate inevitable p-type doping effects of residual dopants (*e.g.*, PMMA, etchant, oxygen, water molecules)^{1,50} which were introduced during transfer process.”

Page 5 in the manuscript,

“The spectrum of the pristine graphene showed a very small D-band ($\sim 1350\text{ cm}^{-1}$), which indicates that high-quality graphene had been successfully grown and transferred onto the substrate with **few defects**.”

Reviewer #2

Comments:

The authors have well-replied this referee's comments point-by-point. However, here raising another question: Is there any correlation between the PFSA thickness and carrier concentration(or sheet resistance)? Now the suggested thickness is ~3.4 nm. It's

crucial to know this thickness effect on the performance/properties of their conductive film (optical transparency, carrier concentration sheet resistance etc).

Response) We appreciate the reviewer's constructive comments. To investigate the relation between PFSA thickness and their doping effects, we used a VASE Ellipsometer to measure PFSA thickness as a function of the concentration of PFSA (Table R4).

Table R4. PFSA thickness as a function of PFSA concentration

PFSA concentration (wt.%)	0.05	0.1	0.3	0.5	1.0	2.0
PFSA thickness (nm)	2.88	3.41	9.66	19.15	40.18	88.99

In this work, we used 0.1 wt.% PFSA solution to dope the graphene, and a resulting film has thickness of a ~3.4-nm PFSA layer on graphene. To investigate the thickness dependency of PFSA layer on electrical properties (*i.e.*, change of R_{sh} , and WF) of the PFSA-doped graphene, we measured the electrical properties as a function of PFSA thickness (Figure R7). Deposition of a ~2.88-nm-thick PFSA layer on top of graphene reduced its R_{sh} substantially by 73.4% compared to pristine graphene; as PFSA thickness increased, R_{sh} did not decrease further, and almost saturated at ~78% (*i.e.* 22% with respect to the pristine graphene) when the PFSA layer was ~10 nm thick.

We used a SKP-5050 Kelvin probe measurement system to measure the change in WF when graphene was doped using PFSA. In contrast to the R_{sh} , WF did not saturate with increasing PFSA thickness, but increased continuously from ~0.8 eV (2.88 nm) to ~1.2 eV (> 40 nm). The change in R_{sh} suggests that the quantity of the PFSA's -SO₃H acidic group that interacts with the graphene surface and causes p-type doping is saturated at PFSA thickness of ~10 nm. However, the further increase of WF with PFSA thickness even after the saturation of R_{sh} can be attributed to the increased interface dipole caused by the thickened PFSA layer on top

of the graphene. To use PFSA-doped graphene as an anode in optoelectronics, the insulating PFSA layer must be thin enough to effectively inject holes from the graphene anode to the overlying anode. Therefore, we used 3.4 nm as an optimal thickness of PFSA layer to dope the graphene.

Figure R3. (a) Normalized sheet resistance, and (b) Work function changes as a function of PFSA layer thickness on top of graphene

Also, the pristine PFSA also shows electrical conductivity (dried phase) in nature. Did authors also test this sample as a reference?

Response) PFSA is composed of fluorinated alkyls with sulfonate ionic groups, so it has ionic conductivity as the reviewer indicated. However, PFSA is intrinsically an electrical insulator. Considering previous studies [Solid State Ionics 2009, 180, 580-584], PFSA film has conductivity of $\sim 10^{-3} \text{ S cm}^{-1}$ (*i.e.* the $10^3 \text{ } \Omega \cdot \text{cm}$ of resistivity), which comes only from ionic conductivity. Since the PFSA is electrically insulating, the R_{sh} we measured from the sample is beyond our measurement range in our setup, which means that PFSA is highly insulating.

Reviewers' comments:

Reviewer #2 (Remarks to the Author):

The authors have prepared the experimental data to support and answer this question. This referee believes the claims are better convincing.

As for the relation between PFSA thickness and R_s , the PFSA is the electrical insulator in nature, thus there should exist an optimized thickness on graphene for achieving highest electrical conductivity; however, based on present data, the R_s tend to saturate at a thickness of $\sim 10\text{nm}$. The R_s don't increase especially when the thickness reaches to 88nm , this fact seems to contradict the insulating property of PFSA, why?

Response to Reviewer's comments

We appreciate the reviewer's valuable comment. We revised our manuscript to comply with the reviewer, and have prepared point-by-point response, which we present here. The revised part in the manuscript is marked in red.

Reviewer #2

Comments:

The authors have prepared the experimental data to support and answer this question. This referee believes the claims are better convincing.

As for the relation between PFSA thickness and R_s , the PFSA is the electrical insulator in nature, thus there should exist an optimized thickness on graphene for achieving highest electrical conductivity; however, based on present data, the R_s tend to saturate at a thickness of ~10nm. The R_s don't increase especially when the thickness reaches to 88nm, this fact seems to contradict the insulating property of PFSA, why?

Response) We appreciate the reviewer's constructive comments. Because PFSA is electrically insulating, vertical conductivity from graphene (bottom) to PFSA layer (top) will decrease as the PFSA thickness increases. In the manuscript, however, we have not mentioned about the vertical resistance of graphene/ PFSA bilayer film, but a sheet resistance (R_{sh}) of graphene because PFSA have used for chemical doping of graphene electrode. R_{sh} can be measured using a lateral resistance rather than vertical resistance through a 2-dimensional conducting materials. R_{sh} measurement system of PFSA-doped graphene is shown in Figure R1. Four point tips electrically contact with graphene surface below the

PFSA layer, not a surface of PFSA layer. Because PFSA is a polymeric soft material, four point tips can have electrical contacts with graphene even though PFSA is thick (~88 nm). Reduction of R_{sh} is saturated at ~10 nm of PFSA, which means that charge transfer (chemical interaction) from graphene to PFSA layer is saturated at PFSA thickness of ~10 nm. To employ the PFSA-doped graphene as an anode in optoelectronics, we used ~3.4 nm-thick PFSA as a dopant layer to facilitate the hole injection via tunneling. Because PFSA is an insulator, holes cannot go through a thick PFSA layer beyond a tunneling thickness.

Again, PFSA layer is electrically insulating and its acidic proton in $-\text{SO}_3\text{H}$ groups induces the p-type doping of graphene and increases hole concentration in graphene, which increases the electrical conductivity of graphene and it was measured as the R_{sh} (lateral conductivity of graphene).

Figure R1. Schematic drawings of four-point probe measurements of PFSA-doped graphene.

Revised parts in the manuscript)

Page 15 in the manuscript,

“ R_{sh} of graphene electrodes was measured using a 4-point probe combined with a Keithley 2400 source meter. R_{sh} decrease of PFSA-doped graphene saturates at PFSA thickness of ~10 nm, and we used 3.4 nm-thick PFSA layer for chemical doping of graphene.”

REVIEWERS' COMMENTS:

Reviewer #2 (Remarks to the Author):

The authors have responded this question, now the claims are better convincing. I, therefore, suggest publication of this manuscript.